# Recycling of memory B cells between germinal center and lymph node subcapsular sinus supports affinity maturation to antigenic drift

Yang Zhang[1,17], Laura Garcia-Ibanez[1,17], Carolin Ulbricht [2,3], Laurence S. C. Lok [4], Jeremy A. Pike[5,6], Jennifer Mueller-Winkler[7], Thomas W. Dennison[4], John R. Ferdinand[4], Cameron J. M. Burnett [1], Juan C. Yam-Puc [1], Lingling Zhang [1,7], Raul Maqueda Alfaro [1,8], Yousuke Takahama [9], Izumi Ohigashi[10], Geoffrey Brown [8], Tomohiro Kurosaki [11,12], Victor L. J. Tybulewicz [7,13], Antal Rot[14,15,16], Anja E. Hauser[2,3], Menna R. Clatworthy[4] & Kai-Michael Toellner [1✉]

Infection or vaccination leads to the development of germinal centers (GC) where B cells evolve high affinity antigen receptors, eventually producing antibody-forming plasma cells or memory B cells. Here we follow the migratory pathways of B cells emerging from germinal centers ($B_{EM}$) and find that many $B_{EM}$ cells migrate into the lymph node subcapsular sinus (SCS) guided by sphingosine-1-phosphate (S1P). From the SCS, $B_{EM}$ cells may exit the lymph node to enter distant tissues, while some $B_{EM}$ cells interact with and take up antigen from SCS macrophages, followed by CCL21-guided return towards the GC. Disruption of local CCL21 gradients inhibits the recycling of $B_{EM}$ cells and results in less efficient adaption to antigenic variation. Our findings thus suggest that the recycling of antigen variant-specific $B_{EM}$ cells and transport of antigen back to GC may support affinity maturation to antigenic drift.

[1] Institute of Immunology and Immunotherapy, College of Medical and Dental Sciences, University of Birmingham, Birmingham, UK. [2] Department of Rheumatology and Clinical Immunology, Charité - Universitätsmedizin Berlin, corporate member of Freie Universität Berlin and Humboldt-Universität zu Berlin, 10117 Berlin, Germany. [3] Deutsches Rheuma-Forschungszentrum (DRFZ), a Leibniz Institute, Charitéplatz 1, 10117 Berlin, Germany. [4] University of Cambridge Molecular Immunity Unit, MRC Laboratory of Molecular Biology, Cambridge Biomedical Campus, Cambridge, UK. [5] Centre of Membrane Proteins and Receptors (COMPARE), Universities of Birmingham and Nottingham, Birmingham, UK. [6] Institute of Cardiovascular Sciences, College of Medical and Dental Sciences, University of Birmingham, Birmingham, UK. [7] The Francis Crick Institute, London, UK. [8] Department of Cell Biology, Center for Research and Advanced Studies, The National Polytechnic Institute, Cinvestav-IPN, Av. IPN 2508, San Pedro Zacatenco, Gustavo A. Madero, 07360 Mexico City, Mexico. [9] Thymus Biology Section, Experimental Immunology Branch, National Cancer Institute, National Institutes of Health, Bethesda, MD 20892, USA. [10] Division of Experimental Immunology, Institute of Advanced Medical Sciences, University of Tokushima, Tokushima 770-8503, Japan. [11] Laboratory of Lymphocyte Differentiation, WPI Immunology Frontier Research Center, Osaka University, Osaka 565-0871, Japan. [12] Laboratory of Lymphocyte Differentiation, RIKEN Center for Integrative Medical Sciences (IMS), Yokohama, Kanagawa 230-0045, Japan. [13] Imperial College, London W12 0NN, UK. [14] Centre for Microvascular Research, The William Harvey Research Institute, Queen Mary University London, EC1M 6BQ London, UK. [15] Centre for Inflammation and Therapeutic Innovation, Queen Mary University London, EC1M 6BQ London, UK. [16] Institute for Cardiovascular Prevention, Ludwig-Maximilians University, 80336 Munich, Germany. [17] These authors contributed equally: Yang Zhang, Laura Garcia-Ibanez. ✉email: k.m.toellner@bham.ac.uk

The hallmark of adaptive immunity is memory, which is mediated by the expansion and long-term survival of antigen-specific lymphocytes, affinity maturation of B lymphocytes, and the long-term production of neutralizing antibody. Affinity maturation of B cells occurs via molecular evolution in germinal centers (GCs)[1]. This involves cycles of B cell proliferation and the mutation of B cell receptor genes, followed by natural selection of B cells expressing the highest affinity B cell receptors. The outputs of the GC reaction are high-affinity antibody-producing plasma cells and memory B cells, both providing long-term immunity[2–5].

Plasma cells can be very long-lived[6], as are memory B cells[7,8]. Interestingly, the affinity-dependent selection of memory B cells in the GC is less stringent than that seen for plasma cells, resulting in a highly variable pool of antigen-specific cells[9–11]. As long-term immunity can be provided by long-lived plasma cells, the advantage of a low-quality B cell output from the GC is not immediately obvious[12]. However, their high variability may provide a pool of cells with the potential to protect against pathogen variants. Memory B cells can sense specific antigen, rapidly enter secondary responses, immediately present antigen to memory T cells[13,14], and generate new plasma cells within days[15–17].

Lymph nodes are important sites for the initiation of the adaptive immune response. They represent a platform where immunological information is sequestered and exchanged. Resident cells, including B cells, occupy distinct anatomical niches, and their movement between different areas of the lymph node is required for the progression of a GC reaction[18]. One important structure in this regard is the subcapsular sinus (SCS), the primary area into which tissue-derived lymph fluid drains, bringing antigens and pathogens. The SCS houses a subset of $CD169^+$ macrophages that are specialized for antigen acquisition and pathogen defense[19] and shuttle antigen to naïve and memory B cells[16,20–22].

Here we follow migration of GC-derived memory-like B cells and show that they enter the SCS. From here they may move on to other lymphoid tissues or interact with SCS macrophages and recycle back to the GC. We show that the migration to the SCS and back is organized S1P and CCL21, respectively, and present data supporting the hypothesis that memory B cell recycling may be a mechanism to survey for and adapt to antigen variants.

## Results

### Appearance of memory-like B cells entering the SCS guided by S1PR.
In order to track the migration of antigen-specific B cells and plasma cells as they emerged from primary GCs in draining lymph nodes (drLN) following immunization, we adoptively transferred 4-hydroxynitrohpenyl (NP)-specific B cells from B1-8i mice[23], which express eGFP under the control of the Prdm1 promoter[24] (labeling plasmablasts and plasma cells with eGFP) and Cdt1-mKO2 hybrid protein[25] (labeling cells in G0/G1 phase of cell cycle with mKO2), and immunized with NP coupled to the carrier protein chicken gamma globulin (CGG). As we previously described[4], plasmablasts emerged from the interface between the GC dark zone and T cell areas (Fig. 1a). Large numbers of antigen-specific B cells were located in the outer follicle surrounding GCs, typically close to the LN SCS (Fig. 1a). Cdt1-mKO2 labeling of these B cells suggests that they were recently activated B cells that emerged from adjacent GCs ($B_{EM}$). This is reminiscent of historical observations describing the accumulation of marginal zone-like memory B cells under the SCS[26,27], and recent descriptions of switched memory B cells in follicles around GCs and under the SCS[16,28].

In order to identify GC-derived $B_{EM}$ in drLNs and their migration to non-reactive distant lymphoid tissues (distLNs), we used the well-established Cγ1-Cre reporter strain, which induces

constitutive expression of GFP in B cells after T cell-dependent activation, which includes GC-lineage B cells[29]. We crossed these with mice expressing B cell receptors specific for the hapten 4-hydroxynitrophenyl (NP)[30,31] and a Cre-inducible eGFP reporter (QM Cγ1Cre mTmG mice)[29,32]. We immunized wild type (WT) host mice that had received a small number of antigen-specific B cells from QM Cγ1Cre mTmG. We observed that GC B cells ($eGFP^+NP^+CD38^{low}Fas^+$) were detectable from 4 d after immunization with maximum numbers seen at 6–10 d (Fig. 1b–d). Within a day of GCs reaching maximum size, there was the emergence of a population of B cells that were $eGFP^+$, NP-binding, $CD38^{high}$, $Fas^{int}$, $CD138^-$, $Bcl6^{low}$ (Fig. 1b) in the drLN (Fig. 1c, d). CD38 is expressed when GC B cells acquire a memory-like phenotype[5]. These cells also started to express other markers associated with memory B cells such as CD73, CD80, and PD-L2[3]. At the same time, antigen-activated B cells were observed in distant lymphoid tissues (Fig. 1b–d). These $eGFP^+$ circulating memory B cells ($B_{CM}$) were confirmed to be antigen-specific, expressed CD62L at similar levels to naïve B cells, and high levels of CD73, CD80, and PDL2 (Fig. 1d). This suggests that the presence of $B_{EM}$ close to the SCS at the peak of the GC response is related to emigration of antigen-activated B cells from the drLN through the SCS, generating systemic cellular B cell immunity.

Further immunohistological examination of drLNs around the peak of $B_{EM}$ migration (Fig. 1e–h) showed that the $eGFP^+$ $B_{EM}$ in B cell follicles surrounding the GC were still in cell cycle (Fig. 1g). Staining with Lyve-1 and ER-TR7, to identify the SCS floor and ceiling respectively, showed that indeed some $B_{EM}$ had moved into the SCS (Fig. 1h). These data suggest that $B_{EM}$ move from the GC into the SCS, from where they may join the efferent lymph flow[33], leaving the drLN to disseminate via blood into distant lymphoid tissues.

Intravital imaging of drLN of Cγ1Cre mTmG mice confirmed that a large number of $eGFP^+$ $B_{EM}$ had actively migrated between the GC and the SCS (Fig. 2a, Suppl. movie 1). $B_{EM}$ entered the SCS lumen (Fig. 2b), where some moved along the SCS (Fig. 2c) presumably migrating towards efferent lymphatics. Surprisingly, some $B_{EM}$, after a short pause around macrophages in the SCS, recycled into the LN follicles through the SCS floor and migrated back towards the GC (Fig. 2b, d, Suppl. Fig. 1, Suppl. movie 2).

To examine the factors that regulate the migration of $B_{EM}$ from GCs, we performed RNA-Seq analysis of FACS sorted $eGFP^+$ $B220^+$ $CD38^+$ $CD95^{int/+}$ $CD138^-$ $B_{EM}$ in LNs comparing them to $eGFP^+$ $B220^+$ $CD38^-$ $CD95^+$ $CD138^-$ GC B cells, and $eGFP^+$ $B220^+$ $CD38^+$ $CD138^-$ $B_{CM}$ and $eGFP^-$ $dTomato^-$ $B220^+$ $CD38^+$ $CD138^-$ naïve B cells from distLNs. Principle component analysis of all genes expressed by these four subsets of cells confirmed a close relationship of $eGFP^+$ $B_{EM}$ with GC B cells, whereas $eGFP^+$ $B_{CM}$ in distLN are much closer to naïve B cells (Fig. 2e). This was also evident in the number of individual genes differentially expressed, with a greater number of genes differentially expressed regarding the transition from naïve to GC B cells, and a larger overlap in genes co-expressed in GC B cells and $B_{EM}$ (Fig. 2f). Analysis of migratory receptors during the transition from GC B cells into $B_{EM}$ by qRT-PCR revealed a loss of expression chemokine receptors known to be associated with B cells location in the GC (Cxcr5, Cxcr4, S1pr2)[34], and increased expression of the receptors S1pr1, S1pr3, S1pr4, Ebi2, Cxcr3, Ccr6, and Ccr7 (Fig. 2g, h, Suppl. Fig. 3). CCR6, EBI2, and CXCR3 are known to be expressed on memory B cells[10,35]. Blockade or deletion of these receptors, however, did not lead to a noticeable change in the appearance of $B_{CM}$ in distLNs (Suppl. Fig. 4a–c). S1P receptors, particularly S1PR1 and S1PR2, are known to direct the location of B cells in the follicle center and their emigration into lymph vessels[34,36]. In vivo S1PR blockade using FTY720 led to a dramatic reduction of $B_{CM}$ in blood and distLNs (Fig. 2i,

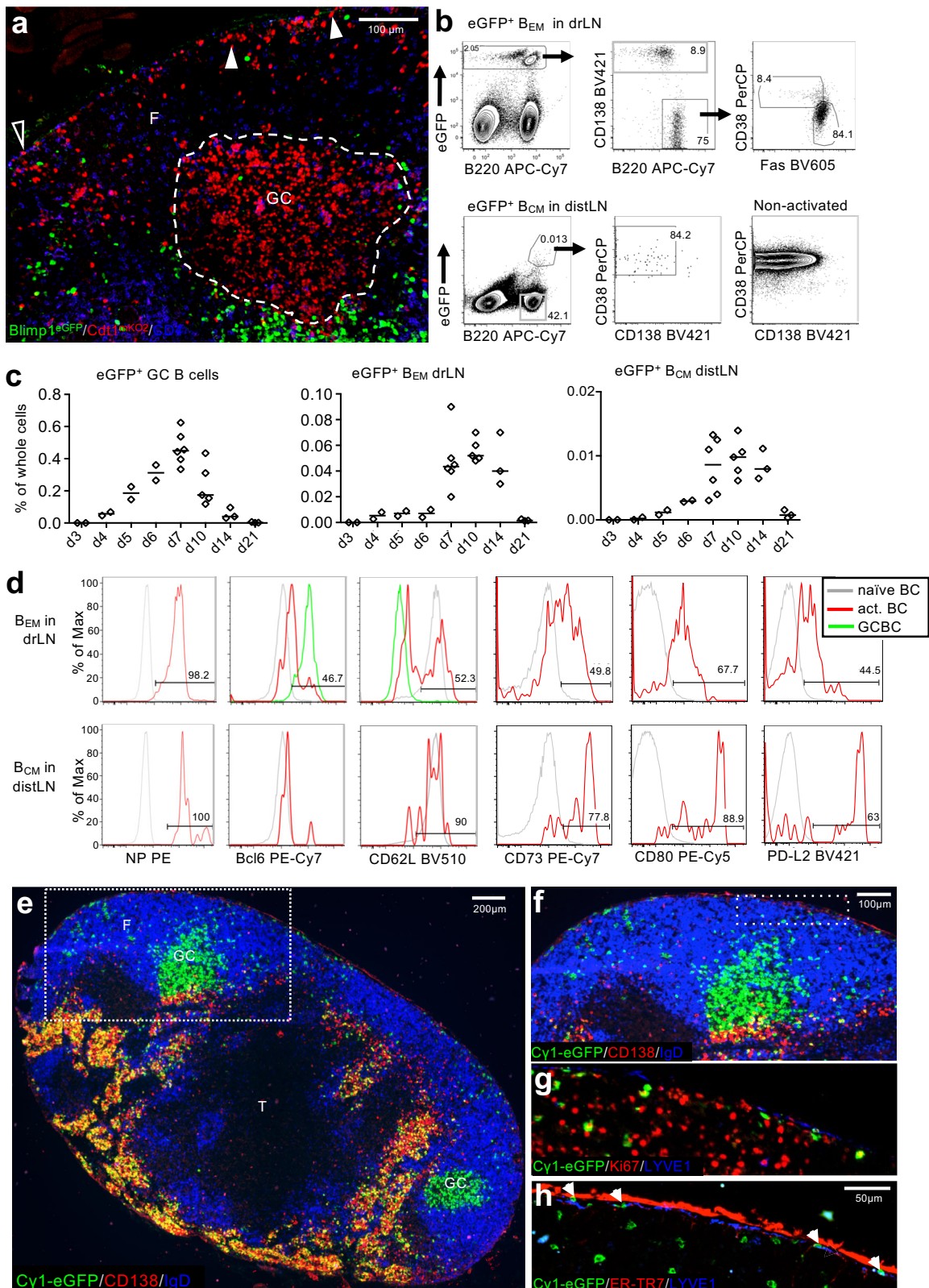

Suppl. Fig. 4d), while there was no noticeable effect on the numbers of other lymphocytes in distant lymphoid tissues (Suppl. Fig. 4d). This suggests that S1PR guides memory B cell migration into the SCS and to lymphatic vessels.

**CCR7 dependent recycling of $B_{EM}$.** The intravital imaging we performed (Fig. 2a–d) showed that many $B_{EM}$ after entering the SCS returned to the follicles. Dendritic cells (DC), arriving in the SCS from afferent lymph, migrate into the lymph node guided by local CCL21 gradients that are sensed by CCR7 on DC[37]. As B cells upregulate CCR7 during the transition from GC B cell to $B_{EM}$ (Figs. 2H, 3A), we hypothesized that a local CCL21 gradient might have a similar role for $B_{EM}$ return into the drLN, as it has for DCs. While CCL21 has not been observed in the subcapsular

**Fig. 1 Appearance of antigen-activated memory-like B cells in drLN and distLN. a** Location of B1-8i/k$^{-/-}$/Blimp1$^{GFP}$/Cdt1m$^{KO2}$ B cells in drLN 6 d after immunization. Antigen-specific B cells in the G0/G1 phase of cell cycle (red) inside the GC (dashed line) and in the follicle (F) close to the SCS (arrow heads). Interfollicular region (open triangle). Blimp-1$^+$ PCs are eGFP (green). Hoechst33342-labeled naïve T cells (blue). Scale bar: 100 µm. Representative image of 3 lymph nodes. **b** Gating of eGFP$^+$ B$_{EM}$ in drLN and B$_{CM}$ and distLN 8 d after immunization of recipients of NP-specific Cγ1Cre QM mTmG B cells. **c** Kinetic of eGFP$^+$ B cell appearance in drLN and distLN. Data merged from two independent experiments ($n = 2$–3). Each datapoint represents one animal. **d** Memory B cell typical markers on B$_{EM}$ and B$_{CM}$ in drLN and distLN. **e** drLN of recipient of QM Cγ1Cre mTmG cells 8 d after immunization. T zone (T). **f** Enlargement of box in **e** showing the eGFP$^+$ B$_{EM}$ close to the subcapsular sinus, **g** Ki-67 expression in B$_{EM}$, **h** same area showing B$_{EM}$ location below the LYVE1$^+$ ER-TR7$^{-ve}$ SCS floor endothelium and inside the SCS (arrowheads). Image is a representative of three lymph nodes.

sinus region surrounding follicles of nonimmunized lymph nodes[37,38], Ccl19 and Ccl21 expression have been shown in immunized lymph nodes in marginal reticular cells (MRC) located under the SCS[39]. In order to test whether CCL21 is expressed in this area, we screened for tdTomato in drLN of Ccl21$^{tdTom}$ mice 8 d after foot immunization. This showed substantial Ccl21 gene expression in stroma located close to the SCS of reactive follicles and in the SCS floor (Fig. 3b, Suppl. Fig. 5a). Immunohistology for CCL21 protein produced a similar pattern, with weak staining for CCL21 around the SCS floor close to follicles containing GCs (Suppl. Fig. 5b).

In order to test how CCR7 ligands affect B$_{EM}$ migration, QM CCR7$^{+/+}$ mT$^+$ and QM CCR7$^{-/-}$ eYFP$^+$ B cells were co-transferred into WT mice and their migration was assessed after immunization. CCR7 is required for the initial activation of naïve B cells, enabling B cell migration into T cell zones[40,41]. Therefore, Ccr7-deficient B cells were underrepresented in activated B cell populations (antigen-specific GC B cells, plasma cells, and B$_{EM}$) in the drLN (Fig. 3c). Despite this, there was an increase in Ccr7$^{-/-}$ B$_{CM}$ in blood, distLNs, spleen, and bone marrow (Fig. 3c, d). This is compatible with a role for CCL21 in orchestrating B$_{EM}$ re-entry into the follicle from the SCS. Without cues sensed by CCR7, B$_{EM}$ is unable to move back from the SCS into the LN parenchyma and therefore appears in larger numbers in blood and distant lymphoid tissues. While CCR7 is an important factor directing the entry of lymphocytes from blood into distant lymphoid tissues, CXCR4 and CXCR5 can provide similar functions[42]. Both are expressed at elevated levels in B$_{CM}$ (Fig. 1, Suppl. Fig. 3).

The non-signaling atypical chemokine receptor 4 (ACKR4) is expressed in the SCS ceiling endothelium and shares the ligands CCL19 and CCL21 with CCR7. To test whether CCR7-mediated retention of B$_{EM}$ in the drLN is dependent on an ACKR4-generated chemokine gradient, we co-transferred QM mT$^+$ Ackr4$^{+/+}$ and QM eYFP$^+$ Ackr4$^{-/-}$ B cells into Ackr4$^{+/+}$ or Ackr4$^{-/-}$ hosts and immunized with NP-CGG. While ACKR4-deficiency on B cells had no significant effect on the size of the GC compartment nor affected B$_{EM}$ numbers in the drLN, ACKR4-deficiency of the LN environment led to decreased numbers of antigen-specific B$_{EM}$ being retained in the drLN and higher numbers appearing in the blood (Fig. 4a). This suggests that chemotactic cues generated in the SCS environment organize B$_{EM}$ reentry into the drLN. In the absence of these, B$_{EM}$ leave the SCS in larger numbers to appear as B$_{CM}$ in the efferent lymph and distLNs.

To further test this, we followed the migration of B$_{EM}$ in the SCS of drLN by fluorescence microscopy in drLNs of Cγ1Cre mTmG mice that were Ackr4$^{+/+}$ or Ackr4$^{-/-}$. This revealed a significantly increased retention of B$_{EM}$ in the SCS of ACKR4$^{-/-}$ drLN 8 d post immunization when B$_{EM}$ recycle (Fig. 4b, c). There was also an increased number of B$_{CM}$ arriving in ACKR4$^{-/-}$ distLNs, and this difference persisted until d 14 (Fig. 4d). Intravital two-photon microscopy confirmed the increased numbers of B$_{EM}$ in the SCS of drLNs of ACKR4-deficient mice (Fig. 4e, f, Suppl. Fig. 6, Suppl. movie 3), and a significantly

reduced number of tracks of B$_{EM}$ recycling into the drLN when ACKR4 was absent (Fig. 4g, Suppl. movie 4).

**B$_{EM}$ recycling supports adaption to antigenic drift.** We next considered the functional significance of B$_{EM}$ LN re-entry from the SCS into the B cell follicle. Of note, we observed that B$_{EM}$ appeared to undergo prolonged interaction with CD169-positive SCS macrophages (Fig. 5a, Suppl. Fig. 7, Suppl. movie 5) before reentering the LN parenchyma. Intravital imaging of cytoplasmic calcium showed an increase in calcium specifically in B cells contacting SCS macrophages (Fig. 5b, Suppl. Fig. 8, Suppl. movie 6), suggesting an antigen-specific interaction between B$_{EM}$ and antigen-carrying SCS macrophages. SCS macrophages are known to transfer antigens to naïve B cells[20,21,43]. A recent study showed similar interactions of antigen-specific memory B cells during secondary responses[16]. For some B$_{EM}$, we observed that they acquired CD169-labeled material from SCS macrophages (Fig. 5c, Suppl. movie 7), suggesting that B$_{EM}$ can acquire and transport antigen from SCS macrophages into the GC. To test this, mice were immunized with rabbit-IgG and eight days later injected with AlexaFluor647-labeled mouse anti-rabbit immune complex (IC) into the foot. Within 10 min, IC was seen associated with SCS macrophages of drLNs. IC was also present inside intranodal lymphatics and entering the lymph node parenchyma. B$_{EM}$ within the SCS were in intimate contact with IC-carrying cells, whereas inside the LN parenchyma, those B$_{EM}$ that were close to the SCS carried speckles of IC (Fig. 5d). Flow cytometry confirmed that 20–30% of B$_{EM}$ carried increased amounts of IC within minutes of IC injection (Fig. 5e, Suppl. Fig. 9)[44].

While the majority of B cells with a history of Cγ1Cre expression is GC-derived, Ig class switch recombination is induced before GCs develop[15,45] and some memory B cells can develop before GCs appear[3]. To confirm that B$_{EM}$ seen recycling are GC derived, we made use of mice with tamoxifen-inducible Cre expressed under the S1pr2 promotor[9] crossed onto Ai14 mice with Cre-inducible tdTomato expression (S1pr2$^{CreERT2}$ Ai14). These mice were foot immunized and treated with tamoxifen 2 d before drLNs were removed at day 8. Light sheet microscopy of live drLNs showed large numbers of tdTomato$^+$ B$_{EM}$ in follicles between GC and SCS (Fig. 5f, Suppl. movie 8). When these mice had been injected with immune complex 10 min before drLNs were taken, immune complexes could be seen in close association with B$_{EM}$ (Fig. 5g, Suppl. movies 9, 10). Taken together, these data suggest that B$_{EM}$ can be activated by specific antigen in the SCS, and then may transport this back towards the GC.

GCs typically contain large amounts of antigen deposited on follicular dendritic cells. Therefore, additional antigen deposition by B$_{EM}$ seems unnecessary, unless the antigen is changing during the course of an infection. B$_{EM}$, are a GC output with highly variable affinity and specificity for antigen, and would therefore include cells that may interact with antigenic variants. To test the hypothesis that B$_{EM}$ recycling has a role in adaption to antigenic drift, we used variants of the hapten NP and measured the adaption of affinity maturation to these variants. The B cell response of C57BL/6 mice is dominated by a canonical IgH VDJ BCR

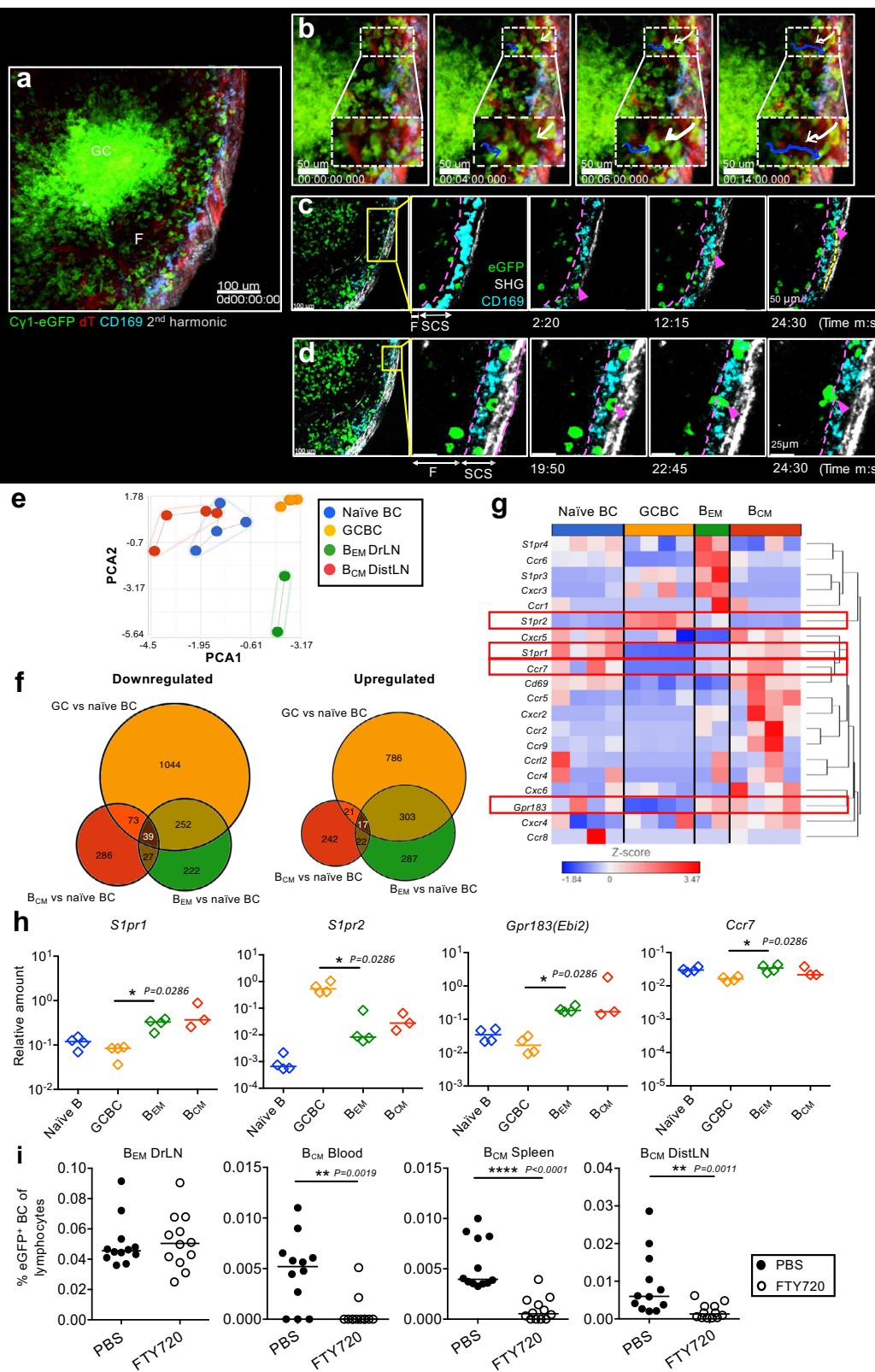

combination that has natural affinity to 4-hydroxy-iodo-phenyl (NIP), and reduced affinity to the variants NP, dinitrophenyl (DNP) and trinitrophenyl (TNP) (Fig. 6a). ACKR4$^{+/-}$ mice were immunized with NIP-KLH. After the onset of B$_{EM}$ recycling, we rechallenged in the same foot with NP, DNP, followed by TNP-KLH. Three days after the last injection we observed a shift in antibody affinity towards TNP (Suppl. Fig. 10). In order to test

whether this was dependent on B$_{EM}$ recycling, the experiment was repeated in Ackr4$^{-/-}$ mice, where B$_{EM}$ cannot undergo recycling. This showed that without B$_{EM}$ recycling, the drift toward the new antigenic variant was significantly reduced (Fig. 6b). This occurred despite ACKR4 deficiency itself not affecting GC sizes, antibody generation, or affinity maturation to immunization when there was no antigenic drift (Suppl. Fig. 11).

**Fig. 2 B$_{EM}$ movement in the drLN. a** Intravital observation of popliteal lymph node from Cγ1Cre mTmG mice 8 d after NP-CGG foot immunization (see suppl. movie 1). **b** Still images showing eGFP$^+$ B$_{EM}$ entering the SCS (blue arrow) and reentering the lymph node follicle from the SCS (white arrow). **c** Images showing an eGFP$^+$ B cells migrating along the SCS. **d** Images showing a B$_{EM}$ reentering the lymph node follicle. Representative of 7 Cγ1Cre mTmG mice ($n = 7$). RNA-Seq data from B cell populations sorted from drLN 8 d after immunization of recipients of Cγ1-Cre QM mTmG B cells. **e** Principal component analysis of global gene expression in sorted populations, **f** Numbers of differentially expressed genes. **g** Heat map of selected chemotactic receptors among mRNA. Genes were hierarchically clustered by Euclidean distance measure. Data are from two independent experiments with four samples (each sample from pooled popliteal lymph nodes from three mice). **h** S1pr1, S1pr2, Ebi2, and Ccr7 mRNA expression analyzed by qRT-PCR. Each diamond represents pooled lymph nodes from four mice. All values are relative to B2m mRNA. Two-tailed Mann-Whitney test, *: $p = 0.0286$. Data representative of three independent experiments (total $n = 12$). **i** B$_{EM}$ in drLN and B$_{CM}$ in blood, spleen, and distLN 8 d after immunization. Mice received FTY720 over 2 d before collection. Each dot represents one mouse, data merged from three independent experiments (total $n = 12$). Two-tailed Mann-Whitney test, **: $p < 0.01$, ****: $p < 0.0001$.

## Discussion

We set out to detect the migratory pathways taken by memory B cells generated from GCs. Lymphocytes leaving lymph nodes are thought to move into intranodal lymphatics before leaving via the medulla and efferent lymph. It was therefore surprising to see memory B cells taking the journey in the opposite direction toward the SCS. Along the SCS they are still able to travel to the hilum and join efferent lymph[46], but this is not the short route. The experiments presented here suggest that S1PR directs the migration of memory B cells into the periphery, leading to their appearance in other lymphoid organs, while CCL21 gradients, organized by ACKR4 expressed in the SCS ceiling, direct recycling of B$_{EM}$ cells back into the lymph node. Another ligand of ACKR4, CCL19, may also be involved, but current detection methods are not sensitive enough to confirm this[47,48]. ACKR4 is also expressed by splenic perimarginal sinus cells, where it may have a similar function[49]. We showed that B$_{EM}$ further express Ccr6, Ebi2 and Cxcr3 mRNA. While inhibition of these receptors did not affect the appearance of memory B cells in distant lymph nodes, this does not mean that they are not playing a role in directing migration of B$_{EM}$ to non-lymphoid tissues or within the reactive LN. CCR6/CCL20 interactions are involved in the migration of follicular GC-derived B cells[35] and CCL20 expressed by subcapsular lymphatics can organize the accumulation of innate-like lymphocytes near the SCS[36]. EBI2 is known to organize the movement of activated B cells to the periphery of the lymph node,[50,51] and its ligands are produced by MRC[52].

The B cell subsets analyzed here are defined by their anatomical location and differential gene expression. For the purpose of this study, they were labeled B$_{EM}$ and B$_{CM}$, although, being freshly generated from GCs they are not memory cells in the classical sense. The two subsets behave differently, with one cell type still in active interaction with antigen and able to re-enter affinity maturation, whereas B$_{CM}$ leave the cell cycle and are probably differentiating towards long-term survival and surveillance for antigen. We have not tested whether epigenetic changes have fixed the differential gene expression in these subsets, so do not know whether the cells represent different lineages evolving or mere differentiation stages that are still plastic and under the influence of the more inflammatory or quiescent microenvironments in different lymph nodes.

SCS macrophages are well known to accumulate and present antigen entering the lymph node via afferent lymph[53]. Naïve B cells scan these macrophages and acquire immune complex via complement receptor cells[44]. Specific antigen-BCR interaction leads to efficient activation of naïve B cells and their migration to the T zone–follicle border, where they can interact with Tfh cells[15,20,21,43]. Others have shown that after secondary immunization, memory B cells can interact with SCS macrophages presenting recall antigen, which leads to interaction with local Tfh cells and rapid plasma cell generation[16,28]. In the current study, we tested the migration and output of memory B cells during the

early peak of the GC reaction. While some memory B cells leave the lymph node and appear in other organs, others interact with SCS macrophages. At this stage, we did not notice substantial plasma cell generation, but found B cells that recycled towards the GC.

Memory B cells are not selected for high affinity and have highly variable BCRs[9–11]. As recycling B$_{EM}$ showed signs of BCR signaling and were seen to transport antigen, we assumed that memory B cell recycling may represent a secondary selection step after B$_{EM}$ test their mutated BCR for the first time on antigen arriving outside the GC. Memory B cells expressing BCRs that are able to interact with antigen in the SCS would be allowed to recycle to the GC together with antigen, where they could undergo further affinity maturation. Many pathogens mutate, leading to antigenic drift. For RNA viruses this happens over different seasons or within individual hosts, such as escape mutants developing during HIV infection. For HIV mutation rates have been reported to be as high as 4 bases per kbp per infected cell[54]. Other RNA viruses mutate slower, with mutation rates for SARS-CoV estimated at around 1 bp per genome per week[55]. The antigenic drift experiment suggests that memory B cell recycling is a mechanism for adaptation to antigenic drift inside the host.

Ackr4-deficiency had a noticeable impact on the adaption to antigenic drift, most likely caused by an inhibition of memory B cell recycling. Dendritic cells entering the LN through the SCS are dependent on the same ACKR4-induced chemokine gradients, however, dendritic cells are more important during the initial of antibody responses when T cells need to be primed[37].

A memory B cell pool with highly variable BCRs is likely to contain B$_{EM}$ variants capable of interacting with antigen variants. B$_{EM}$ recycling would therefore not only transport antigen into the GC but, more importantly, would return the genomic information for BCR variants that may enhance affinity maturation towards the antigen variant. This could contribute to the high number of BCR mutations observed in HIV infection[56], which seem to be caused by repeated B cell re-entry into the GC. While this is not able to stop the emergence of HIV variants, for other viruses this process may be an efficient mechanism of keeping mutations at bay.

## Methods

**Mice and immunization.** All procedures on animals were approved by the University of Birmingham Ethics Committee AWERB (Animal Welfare Ethical Review Board) and done under UK Home Office licenses PPL 70/8011 and PP8702596. Host C57BL/6J mice (wild type, WT) were purchased from Harlan laboratories (stock number: 632c57bl/6J). WT or heterozygous gene-deficient control animals were bred in the same colony as gene-deficient animals. All mice were housed in the Biomedical Services Unit (University of Birmingham) under specific pathogen-free conditions/SPF. At the end experiments mice were humanely killed under anesthesia by cervical dislocation or exsanguination. Mice were mixed sexes and used at 8–12 weeks age. Ackr4$^{tm1.1Rjbn}$ (Ackr4$^{−/−}$) mice[57] were a gift from R. Nibbs (University of Glasgow). Cγ1Cre mTmG mice were generated by crossing Ighg1$^{tm1(cre)Cgn}$ [29] (gift from Stefano Casola, Istituto FIRC di Oncologia

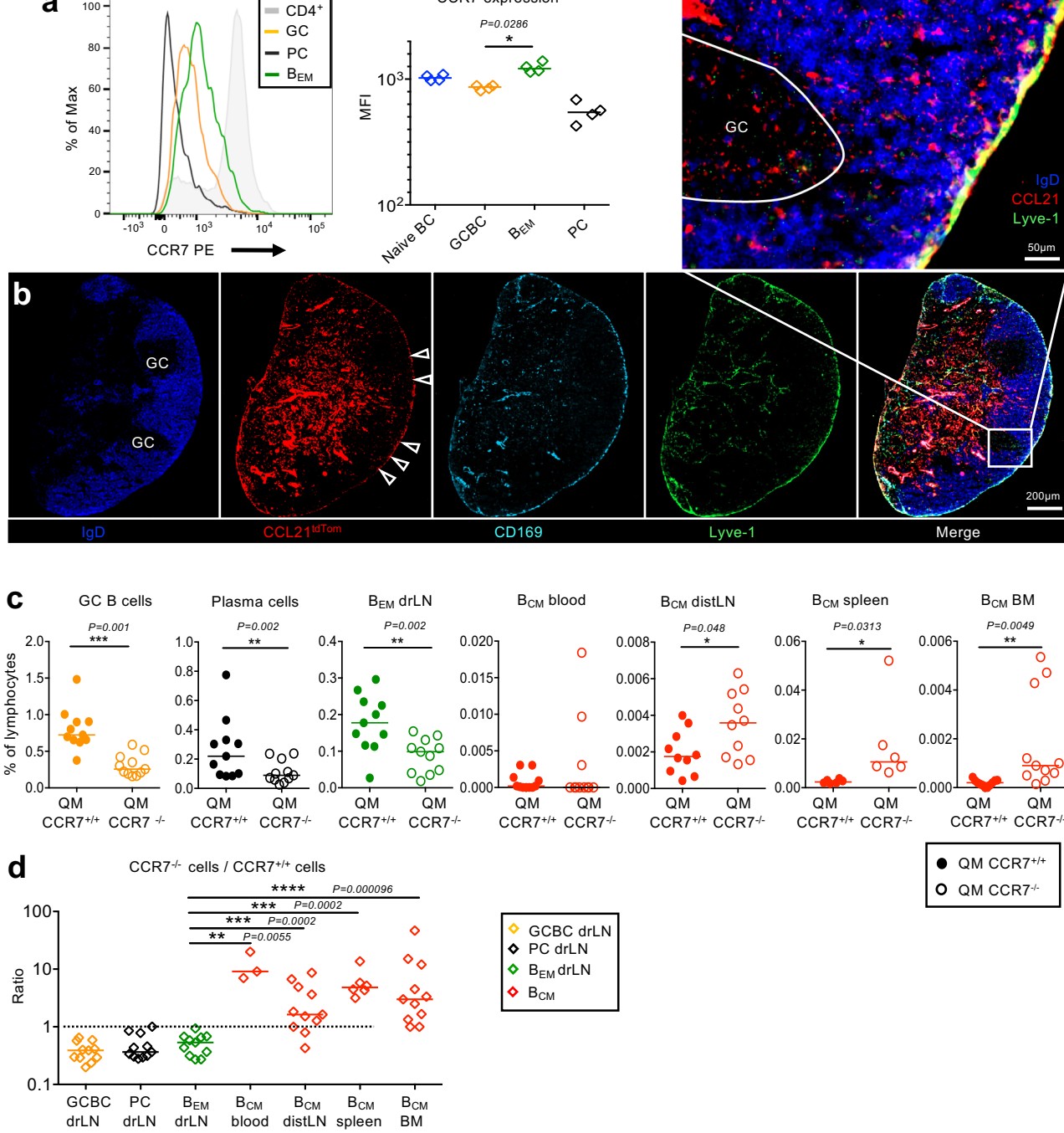

**Fig. 3 CCR7-dependent migration of memory B cells. a** CCR7 expression on different lymphocyte subsets measured by flow cytometry. Each symbol represents one animal. Two-tailed Mann-Whitney test, $^*p$ = 0.0286. **b** drLN from a heterozygous *Ccl21*tdTom/wt reporter mouse 8 d after foot immunization. The SCS ceiling endothelium is missing. Images show dTomato expression in red, and IgD (blue), CD169 (turquoise) and Lyve-1 (green)-specific antibody staining. Arrow heads: Ccl21 expression in SCS. Enlarged box in shows Ccl21a expression under and in the SCS floor endothelium. Image is a representative from 4 lymph nodes. **c** GC B cells, plasma cells (PC), $B_{EM}$ from drLN 8 d after foot immunization of recipients of a mix of QM mT CCR7$^{+/+}$ and QM eYFP CCR7$^{-/-}$ B cells. $B_{CM}$ in blood, distLN, spleen, and bone marrow (BM). Two-tailed Wilcoxon matched-pairs test. **d** Ratio of CCR7$^{-/-}$ to CCR7$^{+/+}$ QM B cells in different tissues. Two-tailed unpaired $t$ test. Each symbol represents one animal. Data are from two independent experiments with 5–6 mice each.

Molecolare, Milan) with Gt(ROSA)26Sor$^{tm4(ACTB-tdTomato, -EGFP)Luo}$ mice[32] (Jackson Laboratory, Stock No. 007914). For all adoptive transfer experiments, variants of QM mice were used, which were homozygous for a NP-specific Ig heavy chain variable region from Igh-J$^{tm1(VDJ-17.2.25)Wabl}$ and Ig kappa light chain deficient (Igk$^{tm1Dhu}$)[30]. Some QM mice contained a constitutively expressed enhanced yellow fluorescent protein (eYFP) derived from Gt(ROSA)26Sor$^{tm1.1(EYFP)Cos}$,[58] (QM eYFP mice), or were crossed onto Gt(ROSA)26Sor$^{tm4(ACTB-tdTomato,-EGFP)Luo}$ (QM mTmG mice). QM CCR7$^{-/-}$ mice are QM crossed with CCR7$^{tm1Rfor}$ mice[59].

*Ccl21*tdTom knock-in mice[38,60] were a kind gift from G Anderson (University of Birmingham). *Ccl21*tdTom heterozygous mice were used to detect Ccl21a gene expression. S1pr2$^{CreERT2}$ Ai14 mice were generated by crossing S1pr2$^{CreERT2}$ [9] onto Gt(ROSA)26Sor$^{tm14(CAG-tdTomato)Hze}$ mice[61] (Jackson Laboratory, Stock No. 007908) to get S1pr2$^{CreERT2}$ Ai14 mice.

NP (4-hydroxy-3-nitrophenyl acetyl) was conjugated to CGG (Chicken γ-globulin) at a ratio of NP$_{18}$-CGG. Mice were immunized into the plantar surface of their rear feet with 20 μg alum precipitated NP$_{18}$-CGG plus $10^5$ chemically

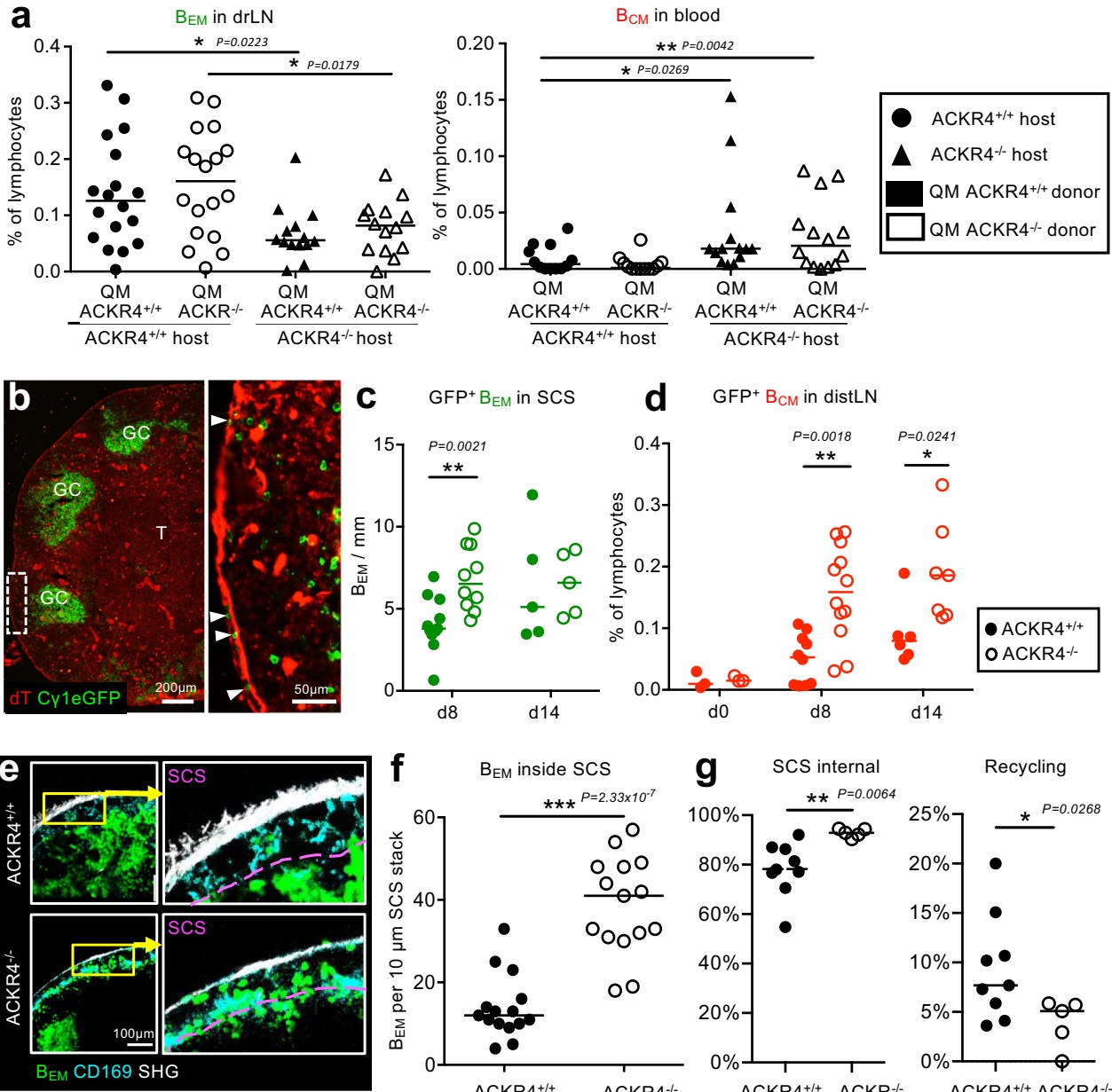

**Fig. 4 Local ACKR4-generated chemokine gradients at the SCS direct migration of memory B cells. a** Frequency of QM ACKR4$^{+/+}$ and QM ACKR4$^{-/-}$ B$_{EM}$ in drLN or B$_{CM}$ in blood of ACKR4$^{+/+}$ and ACKR4$^{-/-}$ hosts. Data merged from three independent experiments with 4–5 mice each. Two-tailed Mann-Whitney test. **b** drLN from an Cγ1-Cre mTmG mouse 8 d after immunization. Arrows: eGFP$^+$ B$_{EM}$ inside the SCS. Representative image of nine lymph nodes. **c** eGFP$^+$ B$_{EM}$ in the SCS of drLN in Cγ1Cre mTmG ACKR4$^{+/+}$ or ACKR4$^{-/-}$ mice at d8 and d14 after foot immunization. **d** B$_{CM}$ in distLN in ACKR4$^{+/+}$ or ACKR4$^{-/-}$ mice after foot immunization. Data in **c** and **d** are merged from two independent experiments with 3–5 mice each. Two-tailed Mann-Whitney test. **e** Intravital observation of B$_{EM}$ entry into SCS in Cγ1Cre mTmG ACKR4$^{+/+}$ or ACRK4$^{-/-}$ drLN. **f** Quantitation of intravital eGFP$^+$ B$_{EM}$ in SCS of ACKR4$^{+/+}$ or ACKR4$^{-/-}$ drLN. Each dot represents cell counts per field of view of a 10 μm Z stack. Pooled data from imaging of three ACKR4$^{+/+}$ and two ACKR4$^{-/-}$ popliteal LNs. Two-tailed unpaired $t$ test, ***$P = 2.33 \times 10^{-7}$. **g** Automated B cell tracking showing fraction of B$_{EM}$ tracks that are inside the SCS or recycling from the SCS into the lymph node parenchyma. Images from five ACKR4$^{+/+}$ and 2 ACKR4$^{-/-}$ popliteal LNs were analyzed. Each symbol corresponds to the average fraction of tracks from one video, total $n$ of tracks = 2478. One-tailed unpaired $t$ test, *$p = 0.0268$; **$p = 0.0064$.

inactivated *Bordetella pertussis* (B.p.) (LEE laboratories, BC, USA)[62]. Popliteal lymph nodes draining the injection site (drLN) and distant axillary and brachial lymph nodes (distLN) were analyzed.

S1PR inhibition: Wt mice received adoptive transfer of $2 \times 10^5$ NP$^+$ B220$^+$ cells from QM Cγ1Cre mTmG. One day later they were foot immunized with NP-CGG. FTY720 (Caymanchem, USA) was given at 1 mg/kg body weight *i.p.* 6 and 7 d later. Tissues were analyzed 8 d after immunization.

EBI2 and CXCR3 inhibition: Wt mice were immunized as above, and NIBR189 (Ebi2 antagonist) or NBI7433 (CXCR3 antagonist) (both from Tocris, UK) were injected i.p. at 60 μg per mouse 5 h before being killed at 8 d after immunization.

For immune complex injections, mice were primed with 20 μg rabbit IgG (PRABP01, BioRad UK) alum precipitated with $10^5$ Bordetella pertussis (B.p) in the foot. Eight days later 4 μg of Alexafluor647 (or Alexafluor488) labeled immune complex (IC) was made with 1:1 ratio of Alexafluor647 (or Alexafluor488) conjugated mouse anti-rabbit IgG and rabbit IgG (Jackson Immunoresearch). This was mixed for 30 min before injecting into the foot. Tissues were usually taken 30 min after injection.

For antigenic drift experiments ACKR4$^{-/-}$ mice and littermate ACKR4$^{+/-}$ control mice were primed with 10 μg of NIP-KLH in alum precipitated with $10^5$ B.p. in the rear feet. 8 d later, mice were boosted with 1 μg of soluble NP-KLH,

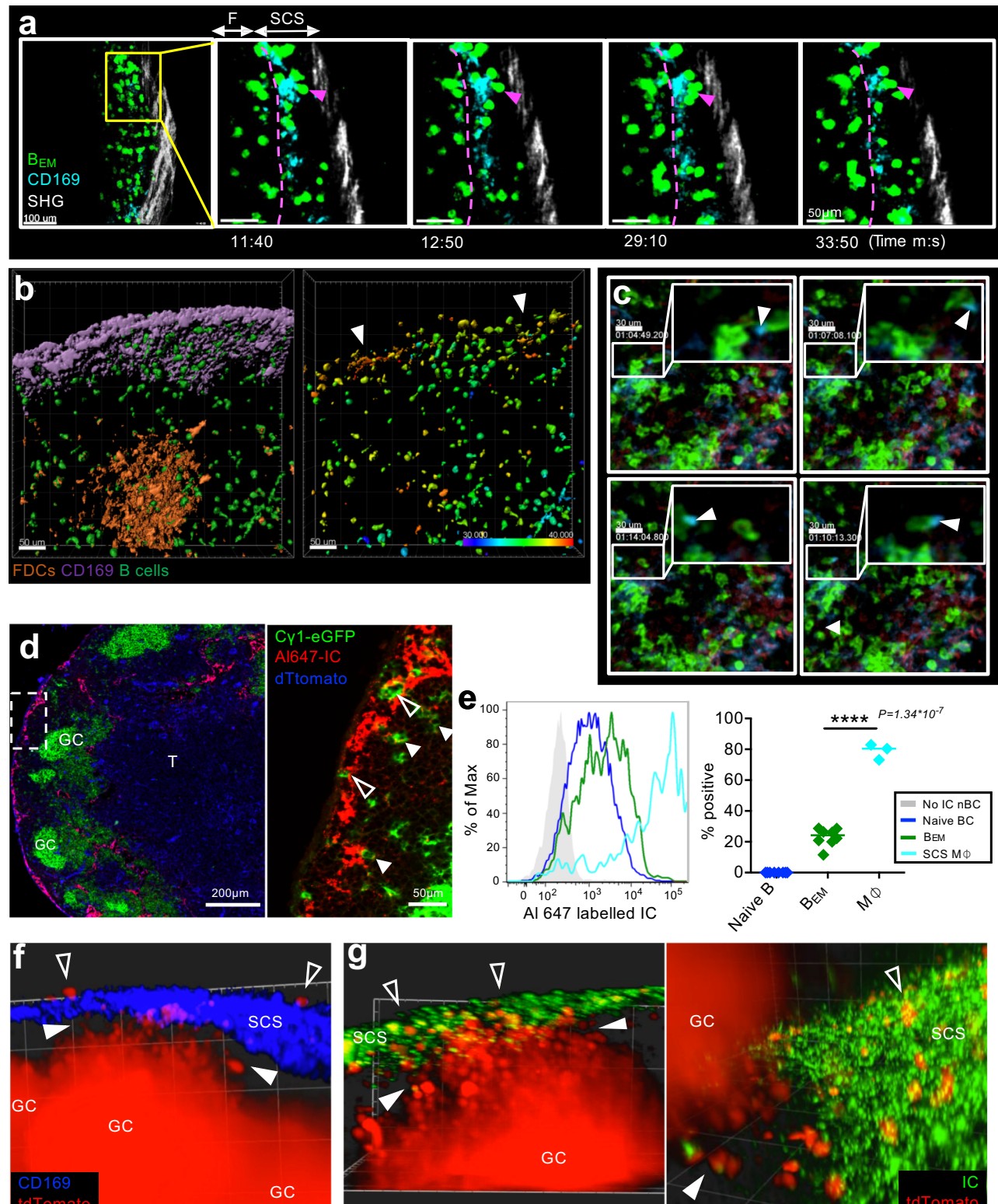

DNP-KLH, and TNP-KLH on the same feet every 2 days (NP conjugates from Biosearch Technologies, USA).

To label early GC derived memory B cells, 4 mg tamoxifen (SIGMA) was given once by gavage to S1pr2^CreERT2 Ai14 mice 6 d after antigen injection into the foot. Popliteal lymph nodes were collected 2 d later.

**Immunohistology.** Lymph node sections were prepared and stained as described previously[31,63]. Frozen sections were cut at 6 μm, and then fixed in acetone. Lymph nodes from *Ccl21*^tdTom heterozygous mouse were fixed in 1% paraformaldehyde overnight followed by 30% sucrose before frozen. Sections were stained with

CD138, IgD, CD3, biotinylated peanut agglutinin (PNA), Ki-67 and LYVE-1, Fibroblasts Specific monoclonal antibody (ER-TR7), CD169, and CCL21. NP-rabbit was house made to detect NP antigen-specific B cells. Secondary antibodies were Cy3-conjugated donkey anti-rat, donkey anti-rabbit, or donkey anti-goat and Alexafluor405 (or Alexafluor488) conjugated streptavidin. Slides were mounted in ProLong Gold antifade reagent (Invitrogen, UK) and left to dry in a dark chamber for 24 h. Images were taken on a Leica DM6000 fluorescence-microscope (Leica Germany), or Zeiss Axio ScanZ1, and Zeiss LSM880 with Airyscan Fast. Image data were processed using Fiji 2.3.0[64], ZEN black (ver. 8.1) or Zen Blue (ver. 1.1.2.0) (Carl Zeiss Germany). All antibodies are listed in Supplementary Tables 1 and 2.

**Fig. 5 B$_{EM}$ interaction with antigen on SCS macrophages. a** Intravital observation of interactions of eGFP$^+$ B$_{EM}$ (green) and SCS macrophages (CD169 turquois). **b** Intravital Ca$^{2+}$ levels in B$_{EM}$. Left: Surface rendering of CD169-stained macrophages (purple) and GC (orange, CD21/35$^{Atto590}$), B1-8$^{hi}$ TN-XXL $^+$ B cells (green). Right: FRET intensity in B$_{EM}$ of same frame with color-coded mean FRET intensity. Arrowheads: clusters of FRET-positive B$_{EM}$ in contact with SCS macrophages. Animation in movie 6. Scale bar 50 μm. **c** Intravital observation (clockwise from top left) of B$_{EM}$ (green) in SCS containing CD169$^+$ material (blue, arrowhead). **d** drLN from an Cγ1Cre mTmG mouse 10 min after foot injection with Alexa647 labeled immune complex (IC). Open Arrow heads: eGFP$^+$ B$_{EM}$ in SCS. Closed arrow heads: Alexa647-IC colocalizing with eGFP$^+$ B$_{EM}$ in follicle. Representative image of three lymph nodes. **e** Left: Alexa647-IC on CD11b$^+$ CD169$^+$ SCS macrophage (green), eGFP$^+$ B$_{EM}$ (red), naïve B cells (blue), or naïve B cells without Alexa647 injection (gray). Right: Percent Alexa647-IC positive B$_{EM}$ or macrophages compared to naïve B cells. Overton subtraction from two independent experiments with three mice each. SCS MΦ: CD169$^+$ subcapsular sinus macrophages. Two-tailed unpaired Student's $t$ test, ****$p = 1.34 \times 10^{-7}$. **f** Light-sheet microscopy of live explanted drLN, showing S1PR2$^{tdTom}$ B$_{EM}$ (red) in located between GC and SCS (closed arrowheads). SCS macrophages (blue). Some B$_{EM}$ are seen in SCS above the SCS layer (open arrowheads). See also movie 8. **g** Same as (**f**) showing Alexa488 labeled IC (green) in the SCS and S1pr2 expression dependent tdTomato positive B$_{EM}$ (red) between GC and SCS (closed arrowheads) or in contact with IC in SCS (open arrowheads). Representative image of four lymph nodes. See also Suppl. movies 9, 10.

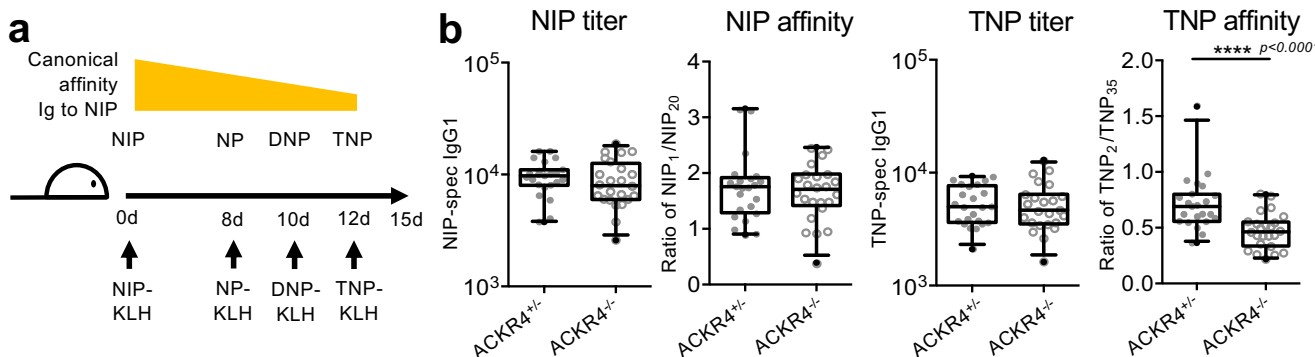

**Fig. 6 B$_{EM}$ interaction with antigen in SCS affects response to antigenic variants. a** Design of antigenic drift experiment using variants of the NIP hapten. **b** NIP-specific and TNP-specific IgG1 antibody titer and affinity in ACKR4$^{+/-}$ and ACKR4$^{-/-}$ mice. Merged data from 4 independent experiments with five mice each (total $n = 20$). Boxes show medians and 50% range, whiskers: 5–95%. Symbols represent individual mice. Two-tailed Mann-Whitney test, ****$p < 0.0001$.

**Flow cytometry and adoptive transfer**. Cells from spleens and lymph nodes were prepared as described[31]. Red blood cells were lysed by ACK lysing buffer (Gibco). Cell suspensions were blocked by CD16/32 diluted in FACS buffer (PBS supplemented with 0.5% BSA plus 2 mM EDTA), followed with staining cocktail. NP was conjugated in house with PE to detect antigen-specific B cells[31]. For detecting transcription factor BCL6, cell suspensions were fixed with BD Cytofix fixation buffer (BD Biosciences) and permeabilized with BD Phosflow Perm buffer III (BD Biosciences), before intracellular staining with Bcl6 PE-Cy7 (BD Biosciences). Samples were analyzed using BD LSRFortessa Analyzer (BD Biosciences, USA) with the software BD FACSDiva (BD Biosciences). Data were analyzed offline with FlowJo (ver. 9 and 10) (FlowJo LLC, USA). All antibodies are listed in Supplementary Tables 1 and 2. All samples were analyzed after gating out dead cells by using live/dead fixable Near-IR dead cell stain kit (Cat: L10119, Invitrogen) or propidium iodide solution (Cat: 431301, BioLegend).

For adoptive transfer experiment, $2 \times 10^5$ NP$^+$ B220$^+$ cells from spleens of fluorescent protein labeled QM background mice were transferred into C57BL6/J hosts 1 d before immunization with NP-CGG in alum on rear feet. In co-transfer experiments, a mix of $1 \times 10^5$ of NP$^+$B220$^+$ B cells of each genotype respectively were injected *i.v.* Gating scheme see Suppl. Fig. 13.

**Cell sorting for qRT-PCR, RNA-seq library preparation, and data analysis**. DrLN and distLN in mice 8 d after foot immunization with NP-CGG in alum and B.p were stained as described above. Naive B cells, GC B cells, plasma cells, B$_{EM}$ cells from drLN and B$_{CM}$ from distLN were sorted using a high-speed cell sorter (MoFlo, Beckman-Coulter). Gating scheme see Suppl. Fig. 12.

For real-time PCR, RNA was purified by using the RNeasy Mini kit (QIAGEN), cDNA preparation was as described as before 31. Real-time PCR from cDNA (qRT-PCR) was done in multiplex with β2-microglobulin and gene expression related to β2-microglobulin gene expression levels. Primers and probes are listed in Supplementary Table 3.

For RNA-seq, cells were directly sorted into 500 μl of Trizol. The total RNA was purified using the RNeasy Plus Micro kit (QIAGEN) according to the manufacturer's instructions. Un-stranded, non-rRNA, non-polyA+ selected libraries were prepared using the SMARTer Ultra Low Input RNA kit for Sequencing v3 (Clontech Laboratories). The libraries were sequenced on the Illumina HiSeq 2000 platform (Illumina, Crick advanced sequencing) as 75 bp paired-end runs. The datasets generated during the current study are available in

the Gene Expression Omnibus (GEO) Database under accession code GSE188687, [https://www.ncbi.nlm.nih.gov/geo/query/acc.cgi?acc=GSE188687].

The sequencing data were analyzed using Partek Flow software, version 8.0.19 (Partek Inc., St. Louis, MO, USA). Paired sequencing data was imported and then aligned to mouse genome GRCm38 (mm10). t-SNE analysis was performed on normalized RNA counts to generate a 2D plot by dimensional reduction. Gene-specific analysis (GSA) tool was used to identify differentially expressed genes against naïve B cells subset as a control. GSA used the lognormal and negative binomial response distribution under the multi-model approach and the lowest maximum coverage of 1.0 was used as the low-value filter. Venn diagrams were produced from differential expression of genes with a log fold change >1 or <−1 and $P$ value < 0.05 using BioVenn[65]. The heatmap was produced from GSA data as described earlier from a predetermined list of genes of interest using the Hierarchical Clustering tool; genes were clustered based on their average Euclidean distance from one another.

**Intravital multiphoton microscopy**. Intravital microscopy of popliteal lymph nodes of Cγ1Cre mTmG mice was performed 8 days after subcutaneous immunization with NP-CGG in alum ppt in the hind hock. Subcapsular sinus macrophages were labeled with CD169-A647 antibody (BioLegend) injected subcutaneously in the hind hock before imaging. The popliteal LN was surgically exposed under a dissecting microscope, and imaged with a Chameleon Vision-S tuneable Ti:Sapphire multiphoton laser and Leica SP8 microscope, with the mouse under inhalational anesthesia and the imaging box kept at 36 °C throughout. Images were acquired using a 25× objective, with one Z stack every 30–40 s, and processed using either Bitplane Imaris 8.2 or Fiji ImageJ 2.3.0[66].

**Automated cell tracking**. Tracking of B$_{EM}$ was performed in 3D using the Fiji 2.3.0 plugin TrackMate[67]. Individual cells were detected using the local maxima of a Laplacian of Gaussian filtered (radius 10 μm) image volume with a quality threshold of 15. A linear assignment problem tacking algorithm was used with maximum frame-to-frame linking distance, maximum tack gap close distance, and maximum track gap closing interval set to 20 μm, 20 μm and two frames, respectively[68]. All tracks less than 5 frames were discarded. A customized Fiji 2.3.0 groovy script was written to automate the tracking and export statistics, available at https://github.com/JeremyPike/SCS-cell-tracking[69].

**Subcapsular sinus segmentation**. The SCS was segmented using a random forest-based pixel classifier implemented and trained using ilastik 1.3.3post3[70]. Four classes were used representing the inner SCS wall, the outer SCS wall, the internal SCS region, and background (no signal). All default ilastik features were used for training which include intensity, edge and texture-based measures over a range of scales. Before training, and prediction, movies were down-sampled by a factor of four in *xy*. Three time-points (selected from either start, middle, or end of movies) were annotated for training. Automated production of training data and prediction was performed using customized ImageJ macros available at https://github.com/JeremyPike/SCS-cell-tracking[69].

**Quantification of interaction between cells and SCS**. All tracking results and SCS segmentations were cropped to 13 z-slices and 32 time-points (for consistency across movies). The ilastik probability maps for individual timepoints were temporally smoothed using a rolling average of three frames. This reduces noise but still allows for movement and drift of the SCS over time. Each voxel was then assigned the class with the maximal probability. For each class (apart from background) all but the largest connected component were discarded. To produce a full SCS segmentation the inner-wall, internal and outer-wall classes were combined. Any holes in the SCS segmentation were filled. The cell tracks were then analyzed with respect to the SCS segmentation, counting occurrences of tracks which entered, left and stayed within the SCS. Entering tracks were defined as starting at least 5 μm from the inner SCS surface and ending within the SCS. Conversely leaving tracks start within the SCS and end at least 5 μm from SCS. Internal tracks start and end within the SCS. This post-processing was performed using Matlab R2020b and a customized script[69].

**Calcium imaging by intravital multiphoton microscopy**. C57BL6/J mice received B cells from B18hi mice (carrying the Vh186.2 heavy chain with high affinity to NP) that contained genetically encoded $Ca^{2+}$ indicator TN-XXL[71] under the control of the CD19 promoter[72]. These were immunized with 10 μg NP-CGG emulsified in complete Freund's adjuvant into the right foot. The popliteal LN was analyzed on day 7. One day prior to imaging, a mixture of 10 μg anti-CD21/35 Fab (clone 7G6) -Atto590 (produced at the DRFZ) for staining follicular dendritic cells and 10 μg CD169-efluor660 (eBioscience) to label SCS macrophages were injected into the footpad.

Intravital two-photon microscopy was performed as described before[73], using a TrimScope II from Lavision Biotec, at an excitation of 850 nm (TiSa) and 1100 nm (OPO). The detection of the fluorescence signals was accomplished with photomultiplier tubes in the ranges of $(466 \pm 20)$ nm, $(525 \pm 25)$ nm, and $(655 \pm 20)$ nm.

TN-XXL is a genetically encoded calcium indicator that consists of a chicken troponin C domain connecting the fluorescent proteins eCFP and Citrine (Suppl. Fig. S8A). These act as a Förster resonance energy transfer (FRET) pair with ECFP as the donor and Citrine as the acceptor fluorophore. Troponin C contains four binding lobes for $Ca^{2+}$ ions. If $Ca^{2+}$ is present or cytosolic concentrations are elevated this leads to a conformation change of the linker peptide that causes donor and acceptor to come into sufficient proximity for FRET emission. When quenched ECFP is excited with one photon at 475 nm, or two photons at 850 nm, citrine will emit fluorescence at 530 nm. If no calcium is present, emission in the blue range of the donor group will be more prominent.

Measurements from six immunized mice were analyzed with image analysis software Imaris 9.2 (Bitplane AG). Raw data were pre-processed using a linear unmixing algorithm[74] to minimize interference of red fluorescence from antibody staining into the green channel of the citrine fluorescence. Relative FRET ratio was calculated by dividing green fluorescence gain by the sum of blue and green fluorescence, and corrected for instrument-specific values and spectral overlap. A colocalization channel was used to measure contact intensity between B cells (citrine-positive, masked on eCFP to exclude OPO influence) and CD169-efluor660 signal. Using the histogram of the colocalization intensity mean of the B cell surfaces, we identified distinguishable populations of B cells (Suppl. Fig. 8F) with either no contact (−) or tight contact (+). All B cells with colocalization intensity of 0 AU were assigned to the (−) group. To choose a threshold value of colocalization intensity for B cells to be assigned to the (+) group, we biexponentially fitted the decay of the histogram and determined the point in which cell numbers intersect the plateau of $y = 9509$ to be 717 AU. Non-contacting B cells and B cells with colocalization intensities >717 AU were filtered and corresponding FRET intensities of all cells at all time points exported for plotting.

**Light-sheet microscopy**. A plane illumination microscope (Zeiss LightSheet Z1) was used to detect memory B cells in popliteal lymph nodes of S1pr2$^{CreERT2}$ Ai14 mice 30 min after injecting AlexFluo488 labeled immune complex at 8 d after immunization with Rabbit IgG on the plantar surface of rear feet. Subcapsular sinus macrophages were labeled with CD169-A647 antibody (BioLegend) injected subcutaneously into the plantar surface of rear foot before imaging. Images were acquired using a 20× Plan-Apochromat objective. Z planes were scaled to 4 μm. The laser wavelengths used were 638 nm, 488 nm, and 561 nm. Images were processed using Vision4D (Arivis).

**Statistical analysis**. All analysis was performed using GraphPad Prism 6 software. To calculate significance two-tailed Mann-Whitney nonparametric test was used. In the experiments where two parameters from the same individual mouse are compared, Wilcoxon matched-pairs signed-rank test (paired nonparametric test) was used to calculate significance. Statistics throughout were performed by comparing data obtained from all independent experiments. *P* values < 0.05 were considered significant (*). *$p < 0.05$, **$p < 0.01$, ***$p < 0.001$, ****$p < 0.0001$. All data are available from the authors upon request.

**Reporting summary**. Further information on research design is available in the Nature Research Reporting Summary linked to this article.

# Data availability

Source data are provided as a Source Data file. The gene expression data have been deposited in the gene expression omnibus (GEO) repository under the accession code GSE188687.

# Code availability

Customized Fiji code and Matlab script for postprocessing of intravital tracking data is available at https://github.com/JeremyPike/SCS-cell-tracking (DOI is https://doi.org/10.5281/zenodo.6351432)[69].

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

## Acknowledgements

Y.Z., C.J.M.B., J.C.Y.P., and K.M.T. were funded by the BBSRC (BB/S003800/1, BB/M025292/1) and MRC (MR/P001874/1). L.G.I and G.B. were supported by EU Marie Curie Initial Training Network DECIDE. V.L.J.T. was supported by the Francis Crick Institute which receives its core funding from Cancer Research UK (FC001194), the UK Medical Research Council (FC001194), and the Wellcome Trust (FC001194). For the purpose of Open Access, the author has applied a CC-BY public copyright licence to any Author Accepted Manuscript version arising from this submission. MRC is supported by a Medical Research Council New Investigator Research Grant (MR/N024907/1), a Chan-Zuckerberg Initiative Human Cell Atlas Technology Development Grant, a Versus Arthritis Cure Challenge Research Grant (21777), and an NIHR Research Professorship (RP-2017-08-ST2-002). A.E.H. was supported by Deutsche Forschungsgemeinschaft

(DFG) TRR130, TP17 and C01, and DFG HA5354/10-1. The authors would like to acknowledge the Flow Cytometry Platform, University of Birmingham for support of flow cytometry experiments.

## Author contributions

Conceptualization: Y.Z., A.R., A.E.H., M.R.C., K.M.T., Methodology: Y.Z., L.G.I., C.U., M.R.C., L.S.C.K., Formal analysis: L.S.C.K., C.J.B., Investigation: Y.Z., L.G.I., C.U., L.S.L., T.W.D., J.R.F., J.M.W., C.J.B., J.Y.P., L.Z., L.S.C.K., R.M.A., Automated tracking and segmentation analysis: J.A.P., Writing: Y.Z., L.G.I., C.U., M.R.C., K.M.T., Provision of gene manipulated animals: Y.T., I.O., T.K., Writing - review and editing: G.B., T.K., V.L.T., A.E.H., M.R.C., K.M.T., Visualization: C.U., L.S.L., T.W.D., J.R.F., Supervision: Y.Z., V.L.T., A.E.H., M.R.C., K.M.T., Funding acquisition: Y.Z., L.S.L., G.B., V.L.T., A.E.H., M.R.C., K.M.T.

## Competing interests

The authors declare no competing interests.
