## [Peer Review File · Nature Communications]

REVIEWER COMMENTS

Reviewer #1 (Remarks to the Author):

This well performed study reveals a number of interesting insights regarding the migration behavior of newly generated memory B cells (Bmem) in lymph nodes (LNs). Using an NP-specific Ig transgenic system, the kinetics of memory B cell emergence in the draining (popliteal) LN, their location in the LN, and their kinetics of appearance in distant (axillary and brachial) LNs is described. Following up on the finding that some drLN Bmem are situated in the subcapsular sinus (SCS) near the germinal centers (GCs), it is observed in excellent quality imaging analysis that some of the cells immediately 'recycle' back into the LN from the sinus. Others are inferred to leave from here into the efferent lymphatic to reach circulation and it is shown that Bmem travel to distLN is inhibited by FTY720, consistent with the expression of S1PR1 by Bmem. Notably, Bmem isolated from the drLN and distLN at the same time point are found to have important differences in their gene expression profiles. This includes high CCR7 expression in the drLN but not distLN Bmem cells. The authors speculate that CCR7 maybe needed for local recycling of Bmem in the drLN and they find that CCR7 KO Bmem travel with increased efficiency to distLN. Examination of mice lacking the CCL19/21 scavenger receptor, ACKR4 (which accumulate CCL21 in the SCS) shows convincingly that Bmem cells accumulate in the SCS when ACKR4 is lacking. They also travel in greater frequencies to distLN. Bmem in or near the SCS are shown to interact with CD169+ SCS macrophages and some of these B cells undergo Ca²⁺ fluxing, consistent with antigen encounter. An example of a cell capturing CD169+ material is shown. Finally, the idea that Bmem recycling through the SCS contributes to supporting a stronger ability to respond to 'antigenic drift' is tested using consecutive immunization with high, intermediate and low affinity antigen for the transgenic BCR. When ACKR4 is lacking, the ability to respond to antigenic drift is compromised. Overall, this study contains a large body of data and is particularly notable for its findings regarding the migration behavior of early Bmem cells, and the evidence for differences between Bmem that remain in the LN versus recirculate. The final parts of the study regarding antigenic drift are less mechanistically definitive, but represent an important effort to explore the logic of early Bmem cell migration behaviors.

Specific comments:

1. For the RNAseq data showing differences between drLN Bmem and disLN Bmem, it would be valuable to know how much of the overall difference is due to the drLN Bmem including cells that are still in cell cycle. How much difference remains once cell cycle signature genes are removed? On that point, how much do the cells in the drLN versus disLN differ in their cell cycle status? Also, is it possible that the differences between drLN and disLN Bmem is reflective of how old the Bmem are rather than these Bmem being separate subsets? As in, the Bmem in the drLN include more newly generated (not yet quiescent) Bmem, whereas those in the disLN were mostly formed longer ago and are likely quiescent. Or perhaps it is that the cells in the disLN have escaped from inflammatory cytokine exposure. These concepts need consideration in the discussion.
2. With the data in Fig 3B regarding CCR7 KO cells, previous work has shown that naïve B cell homing to peripheral LNs is quite dependent on CCR7 (Okada et al., JEM 196, 65, 2002). Do the authors have thoughts on how the CCR7 KO Bmem cells are homing so well to distLN? Are they highly expressing CXCR4? The extent of CCR7 mRNA downmodulation (>100x) in Fig 2H is remarkable. These data do not appear to match what is shown in the heatmap (Fig 2G). Which result is more representative? What are the CCR7 surface levels on disLN Bmem by FACS?
3. The effects of ACKR4 deficiency on Bmem cell accumulation in the SCS are striking. However, a limitation with the use of ACKR4 KO mice that needs to be noted is that it cannot establish whether a particular process is Bmem intrinsic or extrinsic. For example,

while this may not be any more likely, it is possible that the effects on affinity improvement in Fig 4F involve altered migration of Tfh cells or of naïve B cells or DCs. Alternative possibilities such as these need to be considered in the discussion.

4. For the model that CCR7 is guiding Bmem recycling within the drLN, it is inferred that CCR7 ligands are normally available between the SCS and the GC (in the outer follicular mantle). Have the authors attempted to detect CCL21 in this region in WT LNs? CCL21 is not evident here in the WT LN data shown in Ulvmar et al., 2014. If the authors also cannot detect it, perhaps they consider it is below the detection limit, or that CCL19 protein (which is known to be very difficult to detect) is present in this region (although early in situ hybridization data – admittedly of limited sensitivity – did not detect subcapsular signal – Ngo et al., 188, 181, 1998). Perhaps CCL21 entering from peripheral lymphatics is made available on the local subcapsular cells. These possibilities need to be mentioned.

5. The authors note that 'blockade or deletion of CCR6, EBI2 and CXCR3 did not lead to a noticeable change in appearance of Bmem in distLNs'. However, have they examined whether CCR6 contributes to the recycling? Given a report that CCL20 is made by subcapsular lymphatics (Zhang et al., eLife 5, e18156, 2016) this needs to at least be discussed. Similarly, is a role of EBI2 in recycling excluded given the prominent expression of Ch25h by marginal reticular cells (Rodda et al., Immunity 48, 1014, 2018)? Also needs to be mentioned.

6. Are the authors able to provide any quantitative information on how many examples of B cells capturing CD169+ material they observed? If this was only seen very rarely, that should be stated.

7. It should be noted that what are described as macrophages in Fig 4E (assuming these are flow cytometric data) may actually be innate-like lymphocytes that have acquired CD169+ cell membrane material (Gray et al., PlosOne 7, e38238, 2012). Unless microscopy of the isolated cells was done to confirm they are uniformly stained for macrophage markers, the description of the cells should be qualified.

Minor:

1. Pg8 refers to Fig. 5F when it should be Fig 3F.

2. Pg8 in addition to (Carrasco and Batista, 2007; Junt et al., 2007), Phan et al., Nature Immunol 2007 showed naïve B cells capturing antigen from SCS macrophages (both cognately and non-cognately). Additionally, Phan et al., Nature Immunol 2009 showed naïve B cells transporting antigen from SCS cells and transporting it into pre-existing GCs, which is relevant to the current study and might therefore be cited.

3. Suppl Fig S1: the red stromal staining between these samples is very different. At the least some comment about this should be provided in the legend.

4. Overall, the discussion section is too short. As well as highlighting the significant implications of the work, more attention needs to be given to the caveats noted above.

Reviewer #2 (Remarks to the Author):

In this study, Zhang and colleagues explored the possibility that some GC-derived memory B cells recycle back to GC in draining lymph nodes and contribute to affinity maturation to antigen drift. While the claim of "recycling" is interesting, it is far from well supported by the data. In fact, the collection/sequence of observations are not necessarily well connected to conclusively show that a subset of GC-derived memory cells come out of the LZ, go into SCS, acquire antigen from macrophages, and then go back into GC to support further affinity maturation.

Major concerns:

- 1. In Figure 1, the Cg1-Cre reporter would label not only GC-derived cells but also other activated B cells. The authors need to be careful not to equate GFP+ cells with GC-derived cells. The fact that Blimp1-EGFP+ cells show significant accumulation between the T zone and GC DZ does not necessarily mean that these plasmablasts must have come from GC (B1-8hi can give rise to T-dependent but GC-independent plasma cells) or that some plasmablasts do not come out of GC LZ and then quick travel to the T zone-DZ interface through the follicle. Similarly, the appearance of mKO2+ cells around the GC do not necessarily indicate their GC origin. Because the authors are trying to infer kinetic events from static images, these imprecise correspondence makes it really hard to be convinced of the prescribed event sequences.**
- 2. In Figure 1C-D, it seems that some Bcm cells appear 1 day earlier than Bem, and are phenotypically more mature. It is not certain that all Bcm cells in distLN are derived from GC in the reactive LN. It remains possible that some or a majority of those Bcm cells have never gone through the GC reaction.**
- 3. Live imaging (e.g. Fig 2 and related movies) could help establish some of the event sequences. However, their movies, albeit of good quality, are too short to show GC-derived cells arriving into the SCS or coming out of the SCS. There was no quantitative analysis of their observations in any manner, and it is impossible to know how appropriate their preferred interpretation truly is. I would argue, to establish what they want to say, the authors should consider using light-activated marking of their cells.**
- 4. The authors state that "CCR6, EB12, and CXCR3 are known to be expressed on memory B cells (Stoler-Barak et al., 2019; Suan et al., 2017). Blockade or deletion of these receptors, however, did not lead to a noticeable change in the appearance of BCM in distLNs". Where are the supporting data or citation to previous publications?**
- 5. Ackr4 deficiency is unlikely to affect only the migration of Bem cells. The antibody adaptation defect seen in those mice cannot be fully attributed to Bem cells without more carefully controlled experiments.**

Minor points:

On page 6-7, some figures were not correctly cited

Response to Reviewer's Comments

Reviewer #1:

This well performed study reveals a number of interesting insights regarding the migration behavior of newly generated memory B cells (Bmem) in lymph nodes (LNs). Using an NP-specific Ig transgenic system, the kinetics of memory B cell emergence in the draining (popliteal) LN, their location in the LN, and their kinetics of appearance in distant (axillary and brachial) LNs is described. Following up on the finding that some drLN Bmem are situated in the subcapsular sinus (SCS) near the germinal centers (GCs), it is observed in excellent quality imaging analysis that some of the cells immediately 'recycle' back into the LN from the sinus. Others are inferred to leave from here into the efferent lymphatic to reach circulation and it is shown that Bmem travel to distLN is inhibited by FTY720, consistent with the expression of S1PR1 by Bmem. Notably, Bmem isolated from the drLN and distLN at the same time point are found to have important differences in their gene expression profiles. This includes high CCR7 expression in the drLN but not distLN Bmem cells. The authors speculate that CCR7 maybe needed for local recycling of Bmem in the drLN and they find that CCR7 KO Bmem travel with increased efficiency to distLN. Examination of mice lacking the CCL19/21 scavenger receptor, ACKR4 (which accumulate CCL21 in the SCS) shows convincingly that Bmem cells accumulate in the SCS when ACKR4 is lacking. They also travel in greater frequencies to distLN. Bmem in or near the SCS are shown to interact with CD169+ SCS macrophages and some of these B cells undergo Ca²⁺ fluxing, consistent with antigen encounter. An example of a cell capturing CD169+ material is shown. Finally, the idea that Bmem recycling through the SCS contributes to supporting a stronger ability to respond to 'antigenic drift' is tested using consecutive immunization with high, intermediate and low affinity antigen for the transgenic BCR. When ACKR4 is lacking, the ability to respond to antigenic drift is compromised. Overall, this study contains a large body of data and is particularly notable for its findings regarding the migration behavior of early Bmem cells, and the evidence for differences between Bmem that remain in the LN versus recirculate. The final parts of the study regarding antigenic drift are less mechanistically definitive, but represent an important effort to explore the logic of early Bmem cell migration behaviors.

We thank the reviewers for their positive and constructive reviews, which led to substantial improvements of this manuscript.

Specific comments:

1. For the RNAseq data showing differences between drLN Bmem and disLN Bmem, it would be valuable to know how much of the overall difference is due to the drLN Bmem including cells that are still in cell cycle. How much difference remains once cell cycle signature genes are removed? On that point, how much do the cells in the drLN versus disLN differ in their cell cycle status? Also, is it possible that the differences between drLN and disLN Bmem is reflective of how old the Bmem are rather than these Bmem being separate subsets? As in, the Bmem in the drLN include more newly generated (not yet quiescent) Bmem, whereas those in the disLN were mostly formed longer ago and are likely quiescent. Or perhaps it is that the cells in the disLN have escaped from inflammatory cytokine exposure. These concepts need consideration in the discussion.

Data shown in Fig. 1 are from experiments where naïve QM B cells were adoptively transferred one day before immunisation, and the vast majority of memory B cells analysed in Fig 1b, c are newly formed and were generated during the ongoing response. We agree with the reviewer that differences between draining and distant lymph nodes may have developed because the cells have arrived in a non-reactive tissue, or because they have left the GC environment for slightly longer and are "older".

As suggested, we have tested the impact of cell cycle associated or inflammatory response genes on the phenotypes. There is some association in the transition from B_{EM} to B_{CM} with cell cycle related genes and little with inflammatory response genes (Fig. I below and new suppl. Fig. S2). Removing either or both these gene sets from our gene expression data did not affect the separation of these population in the principal component analysis.

We would like to point out that we have not followed these “subsets” over time or have tested epigenetic fixation of gene expression patterns. Therefore, we do not know whether the “subsets” we defined in this study represent long term differentiation patterns or transient differentiation stages, as this was not the aim of this study. We have added these considerations to the revised discussion.

Further, while doing this gene set enrichment analysis, we realized that the transition from GC B cells to B_{EM} fits very well with data published by Laidlaw et al. ¹. This is now shown in new suppl. Fig. S2c.

2. With the data in Fig 3b regarding CCR7 KO cells, previous work has shown that naïve B cell homing to peripheral LNs is quite dependent on CCR7 (Okada et al., JEM 196, 65, 2002). Do the authors have thoughts on how the CCR7 KO Bmem cells are homing so well to distLN? Are they highly expressing CXCR4? The extent of CCR7 mRNA downmodulation (>100x) in Fig 2H is remarkable. These data do not appear to match what is shown in the heatmap (Fig 2G). Which result is more representative? What are the CCR7 surface levels on distLN Bmem by FACS?

Apologies for this error. The original dataset contained plasma cells, which had been removed from the dataset, as they did not add valuable information. Unfortunately, an error was made during revision and the plot in Fig. 2h showed the plasma cell instead of the B_{CM} data. This has been corrected. All other data were correct. The complete dataset is now shown in suppl. Fig. S3.

3. The effects of ACKR4 deficiency on Bmem cell accumulation in the SCS are striking. However, a limitation with the use of ACKR4 KO mice that needs to be noted is that it cannot establish whether a particular process is Bmem intrinsic or extrinsic. For example, while this may not be any more likely, it is possible that the effects on affinity improvement in Fig 4F involve altered migration of Tfh cells or of naïve B cells or DCs. Alternative possibilities such as these need to be considered in the discussion.

This was a concern similarly raised by reviewer #2. We have published that Ack4r4 deficiency affects dendritic cell migration from the SCS in the lymph node parenchyma². We are not aware that Tfh cells do the same journey. In an unpublished study, we performed a large series of experiments to test whether ACKR4 affects affinity maturation and found no changes. Although these data are

supposed to be part of a different study, we present some in the new Fig. S11. Caveats are also mentioned in the new discussion.

4. For the model that CCR7 is guiding Bmem recycling within the drLN, it is inferred that CCR7 ligands are normally available between the SCS and the GC (in the outer follicular mantle). Have the authors attempted to detect CCL21 in this region in WT LNs? CCL21 is not evident here in the WT LN data shown in Ulvmar et al., 2014. If the authors also cannot detect it, perhaps they consider it is below the detection limit, or that CCL19 protein (which is known to be very difficult to detect) is present in this region (although early in situ hybridization data – admittedly of limited sensitivity – did not detect subcapsular signal – Ngo et al., 188, 181, 1998). Perhaps CCL21 entering from peripheral lymphatics is made available on the local subcapsular cells. These possibilities need to be mentioned.

There are a number of studies providing data on CCL19/21 expression in the subcapsular sinus area. Huang et al (Fig. 2)³ show CCL19 and CCL21 expression in MRC of immunised lymph nodes. Others have not found expression, however, this may be due to issues with sensitivity because non-immunised lymph nodes were studied^{2, 4, 5}.

We have tested CCL21 expression in immunised lymph nodes using immunohistology for RFP or by using CCL21^{RFP} reporter mice⁵ to detect gene expression. In these reactive lymph nodes, CCL21 is expressed under the subcapsular sinus surrounding follicles, particular those containing germinal centres. This is shown in new Fig. 3b and suppl. Fig. S5.

We have no data on CCL19 expression, which, as the reviewer correctly states, is difficult to detect and reporter systems do not exist. This has been added to the discussion.

5. The authors note that ‘blockade or deletion of CCR6, EB12 and CXCR3 did not lead to a noticeable change in appearance of Bmem in distLNs’. However, have they examined whether CCR6 contributes to the recycling? Given a report that CCL20 is made by subcapsular lymphatics (Zhang et al., eLife 5, e18156, 2016) this needs to at least be discussed. Similarly, is a role of EB12 in recycling excluded given the prominent expression of Ch25h by marginal reticular cells (Rodda et al., Immunity 48, 1014, 2018)? Also needs to be mentioned.

Thank you for pointing this out. This is all correct. While we were able to test the effects of these different ligands on the appearance of memory B cells in different lymph nodes, it is technically more challenging to follow B_{EM} recycling in presence of absence of these ligands, and we agree that more could be done to fully address this. As suggested, these points have been added to the discussion.

6. Are the authors able to provide any quantitative information on how many examples of B cells capturing CD169+ material they observed? If this was only seen very rarely, that should be stated.

With the complex architecture, the large and complex shape of the macrophages, and the small amount of material transferred, we were not able to accurately quantify these *in vivo* observations. We can, however, confirm that these interactions occur regularly. To demonstrate this, we have added a further supplementary videos (suppl. movie 5), showing several occasions where B cells are seen migrating with macrophage material. Nine such interactions are shown, demonstrating that these events are not rare.

7. It should be noted that what are described as macrophages in Fig 4E (assuming these are flow cytometric data) may actually be innate-like lymphocytes that have acquired CD169+ cell membrane material (Gray et al., PlosOne 7, e38238, 2012). Unless microscopy of the isolated cells was done to

confirm they are uniformly stained for macrophage markers, the description of the cells should be qualified.

Thank you for this comment. We are aware that innate-like lymphocytes can acquire CD169. To exclude these cells, all flow cytometry samples were counterstained for CD11b, CD11c, F4/80 to identify macrophages. The full gating scheme for SCS macrophages is now included as suppl. Fig. S9. Apologies for omitting this important information.

Minor:

1. Pg8 refers to Fig. 5F when it should be Fig 3F.

Corrected

2. Pg8 in addition to (Carrasco and Batista, 2007; Jun et al., 2007), Phan et al., Nature Immunol 2007 showed naïve B cells capturing antigen from SCS macrophages (both cognately and non-cognately). Additionally, Phan et al., Nature Immunol 2009 showed naïve B cells transporting antigen from SCS cells and transporting it into pre-existing GCs, which is relevant to the current study and might therefore be cited.

References have been added

3. Suppl Fig S1: the red stromal staining between these samples is very different. At the least some comment about this should be provided in the legend.

We note the reviewer's concern with this figure (new Fig. S6). For each condition we presented a 10 µm-thick section of the imaged lymph node where the subcapsular sinus can be visualised clearly. Stromal mTomato signal strength changes significantly at different tissue depths, and varies in different lymph nodes during intravital imaging. As suggested we have clarified this in the figure legend.

4. Overall, the discussion section is too short. As well as highlighting the significant implications of the work, more attention needs to be given to the caveats noted above.

Attention has been given to the caveats.

Reviewer #2 (Remarks to the Author):

In this study, Zhang and colleagues explored the possibility that some GC-derived memory B cells recycle back to GC in draining lymph nodes and contribute to affinity maturation to antigen drift. While the claim of "recycling" is interesting, it is far from well supported by the data. In fact, the collection/sequence of observations are not necessarily well connected to conclusively show that a subset of GC-derived memory cells come out of the LZ, go into SCS, acquire antigen from macrophages, and then go back into GC to support further affinity maturation.

Major concerns:

1. In Figure 1, the Cg1-Cre reporter would label not only GC-derived cells but also other activated B cells. The authors need to be careful not to equate GFP+ cells with GC-derived cells. The fact that Blimp1-EGFP+ cells show significant accumulation between the T zone and GC DZ does not necessarily mean that these plasmablasts must have come from GC (B1-8hi can give rise to T-

dependent but GC-independent plasma cells) or that some plasmablasts do not come out of GC LZ and then quick travel to the T zone-DZ interface through the follicle.

Similarly, the appearance of mKO2+ cells around the GC do not necessarily indicate their GC origin. Because the authors are trying to infer kinetic events from static images, these imprecise correspondence makes it really hard to be convinced of the prescribed event sequences.

We thank the reviewer for this comment. Indeed, our own work shows that immunoglobulin class switching and Cg1 germline expression predates germinal centre formation^{6,7,8}. Others have clearly shown that memory B cells can develop independently of germinal centres⁹. At the time of performing this study we had only access to Cg1-Cre and used this as a surrogate marker for GC-derived memory B cell generation, as the majority of cells marked by this reporter are GC-derived.

We have repeated some experiments using the GC-specific S1PR2^{CreERT2} reporter mouse. New Fig. 5f-g and new supplementary movies 8-10 show light sheet microscopy of live lymph nodes with many GC-derived memory B cells between GC and SCS and in the SCS, some in interaction with immune complex.

Additionally, we now show data showing that the gene expression profile of B_{EM} is extremely similar to early GC derived memory B cells described by Laidlaw et al¹ (new Fig. S2c).

Regarding the comment on Fig. 1a:

We have published extensive evidence that Blimp1-eGFP cells in the area between GC and T zone are GC derived¹⁰. Fig. 1a is shown mainly as an introduction to explain background and context and explain tissue architecture. Even if some of these plasma cells would have developed outside the GC, this would be irrelevant for the conclusions of the current study.

mKO2+ cells: This study is not based on “inferring kinetic events from static images”, as we provide several examples of kinetic observations of various events using intravital microscopy (see supplementary movies).

2. In Figure 1C-D, it seems that some Bcm cells appear 1 day earlier than Bem, and are phenotypically more mature. It is not certain that all Bcm cells in distLN are derived from GC in the reactive LN. It remains possible that some or a majority of those Bcm cells have never gone through the GC reaction.

The reviewer overinterprets data in Fig. 1c. This is probably because B_{CM} numbers for distant LNs on day 4 had not been shown. These were omitted because there is no significant increase in B_{CM} numbers by this day. We have now added these data, showing that B_{CM} appear *after* the appearance of B_{EM} and GC in the draining lymph node.

Fig. 1d shows data from the peak of the response at day 8 after immunisation. The fact that B_{CM} cells appearing in distant LN by day 8 are more phenotypically mature than in the draining LN is not surprising, as they have migrated through lymph and blood by this stage.

We agree that some Cg1Cre cells can be non-GC derived, as discussed above.

3. Live imaging (e.g. Fig 2 and related movies) could help establish some of the event sequences. However, their movies, albeit of good quality, are too short to show GC-derived cells arriving into the SCS or coming out of the SCS. There was no quantitative analysis of their observations in any manner, and it is impossible to know how appropriate their preferred interpretation truly is. I would argue, to establish what they want to say, the authors should consider using light-activated marking of their cells.

We have spent considerable time reanalysing our multiphoton microscopy movies. The length of the movies shown is at the limit of what is technically possible. We now provide several examples of tracks of cells making the whole journey from the subcapsular sinus to the GC, and we have added evidence in new suppl. Fig. S1 and new suppl. movie 2.

Furthermore, we analysed the movies using a non-biased and automated 3D cell tracking and subcapsular sinus segmentation image analysis workflow. This customized workflow was implemented using open-source tools and the relevant scripts have been made freely available for full transparency. New Fig. 4g quantifies the numbers of tracks of cells returning from the SCS into the lymph node parenchyma, and clearly indicates a significant reduction of recycling once CCL19/21 gradients are disrupted.

4. The authors state that "CCR6, EBI2, and CXCR3 are known to be expressed on memory B cells (Stoler-Barak et al., 2019; Suan et al., 2017). Blockade or deletion of these receptors, however, did not lead to a noticeable change in the appearance of BCM in distLNs". Where are the supporting data or citation to previous publications?

These data have been added as Suppl. Fig. S4.

5. Ackr4 deficiency is unlikely to affect only the migration of Bem cells. The antibody adaptation defect seen in those mice cannot be fully attributed to Bem cells without more carefully controlled experiments.

This point was also raised by reviewer #1. See reply to reviewer #1, point 3. Data have been added to Fig. S11 and we discuss this point in the discussion.

Minor points:

On page 6-7, some figures were not correctly cited
Apologies. This has been corrected

References:

1. Laidlaw BJ, Schmidt TH, Green JA, Allen CD, Okada T, Cyster JG. The Eph-related tyrosine kinase ligand Ephrin-B1 marks germinal center and memory precursor B cells. *J Exp Med* **214**, 639-649 (2017).
2. Ulvmar MH, et al. The atypical chemokine receptor CCRL1 shapes functional CCL21 gradients in lymph nodes. *Nat Immunol* **15**, 623-630 (2014).
3. Huang HY, et al. Identification of a new subset of lymph node stromal cells involved in regulating plasma cell homeostasis. *Proc Natl Acad Sci U S A* **115**, E6826-E6835 (2018).
4. Ngo VN, Tang HL, Cyster JG. Epstein-Barr virus-induced molecule 1 ligand chemokine is expressed by dendritic cells in lymphoid tissues and strongly attracts naive T cells and activated B cells. *J Exp Med* **188**, 181-191 (1998).

5. Kozai M, *et al.* Essential role of CCL21 in establishment of central self-tolerance in T cells. *J Exp Med* **214**, 1925-1935 (2017).
6. Toellner K-M, Gulbranson-Judge A, Taylor DR, Sze DM-Y, MacLennan ICM. Immunoglobulin switch transcript production in vivo related to the site and time of antigen-specific B cell activation. *J Exp Med* **183**, 2303-2312 (1996).
7. Toellner K-M, *et al.* T helper 1 (Th1) and Th2 characteristics start to develop during T cell priming and are associated with an immediate ability to induce immunoglobulin class switching. *J Exp Med* **187**, 1193-1204 (1998).
8. Roco JA, *et al.* Class-Switch Recombination Occurs Infrequently in Germinal Centers. *Immunity* **51**, 337-350 e337 (2019).
9. Weisel FJ, Zuccarino-Catania GV, Chikina M, Shlomchik MJ. A Temporal Switch in the Germinal Center Determines Differential Output of Memory B and Plasma Cells. *Immunity* **44**, 116-130 (2016).
10. Zhang Y, *et al.* Plasma cell output from germinal centers is regulated by signals from Tfh and stromal cells. *J Exp Med* **215**, 1227-1243 (2018).

REVIEWERS' COMMENTS

Reviewer #1 (Remarks to the Author):

In their revised manuscript the authors have adequately addressed my principal concerns. This includes the addition of substantial amounts of new data.

Minor

I'm not sure that the authors addressed my earlier point 2 regarding how CCR7 KO Bmem cells are homing so efficiently to distLN given the role of CCR7 in B cell movement from blood into LN. Some comment on this should be provided. Perhaps it is mediated by the elevated CXCR4.

Regarding S1P-S1PR1 promoting cell movement from follicles into the SCS, it may reinforce the authors conclusions on this point to cite that this was earlier shown (genetically as well as pharmacologically) for innate like lymphocytes (ref 47). The similar shuttling behavior between follicle and SCS of the innate like lymphocytes (47) and memory B cells might also be commented on in the discussion as it may represent a conserved migration program.

Reviewer #3 (Remarks to the Author):

The authors made considerable and effective efforts in addressing all the concerns raised by the reviewer. I find the study to be of high quality and significant importance. I only have few very minor suggestions:

1. The authors briefly discuss the possible limitations of using ACKR4 KO, highlighting why potential effects mediated by changes in DC migration are unlikely to contribute to adaptation to antigenic drift. While the argument makes perfect sense, it might still be good to expand this point and include the possibility of additional potential mechanisms that may contribute. In particular, as suggested by reviewer 1, it will be good to discuss the potential role of naïve B cells, which can carry antigens from the SCS to FO via complement receptors, and Tfh cells that may 'follow' B cells to these regions in a manner similar to them being 'dragged' by activated B cells into GC sites.

2. Could the authors add a brief explanation of what model they used to highlight stroma cells (e.g., how does the tdTomato signal in S1 targets the stroma)? This information should be added to the methods and legends. In addition, because the tdTomato line is crossed to several different models in this study, it will be helpful to specify what the 'tdTomato' means in the legends of each figure and movie, where it appears.

3. line 141-3: the statement that Cxcr5 and Cxcr4 expression is downregulated is mentioned twice in the same sentence.

Rebuttal to additional comments from reviewers #1 and #3:

Reviewer #1:

- *I'm not sure that the authors addressed my earlier point 2 regarding how CCR7 KO Bmem cells are homing so efficiently to distLN given the role of CCR7 in B cell movement from blood into LN. Some comment on this should be provided. Perhaps it is mediated by the elevated CXCR4.*

Apologies for omitting this important point. Our data in Fig. 2g and suppl. Fig. S3 show elevated levels of CXCR4 (and CXCR5). This is now mentioned with the results on CCR7-/- cell transfers.

- *Regarding S1P-S1PR1 promoting cell movement from follicles into the SCS, it may reinforce the authors conclusions on this point to cite that this was earlier shown (genetically as well as pharmacologically) for innate like lymphocytes (ref 47). The similar shuttling behavior between follicle and SCS of the innate like lymphocytes (47) and memory B cells might also be commented on in the discussion as it may represent a conserved migration program.*

Thank you. We have added this and mention ref 47 (now ref 36) in results and discussion.

Reviewer #3:

- *1. The authors briefly discuss the possible limitations of using ACKR4 KO, highlighting why potential effects mediated by changes in DC migration are unlikely to contribute to adaptation to antigenic drift. While the argument makes perfect sense, it might still be good to expand this point and include the possibility of additional potential mechanisms that may contribute. In particular, as suggested by reviewer 1, it will be good to discuss the potential role of naïve B cells, which can carry antigens from the SCS to FO via complement receptors*

This point (know role of naïve B cells transporting antigen) is already mentioned in the third paragraph of the discussion.

- *, and Tfh cells that may 'follow' B cells to these regions in a manner similar to them being 'dragged' by activated B cells into GC sites.*

We are not sure about this point by the reviewer. We are aware of data of B cells dragging T cells behind within individual microenvironments, e.g. within GCs, but not that B cells drag Tfh cells into different microanatomical compartments.

- *Could the authors add a brief explanation of what model they used to highlight stroma cells (e.g., how does the tdTomato signal in S1 targets the stroma)? This information should be added to the methods and legends. In addition, because the tdTomato line is crossed to several different models in this study, it will be helpful to specify what the 'tdTomato' means in the legends of each figure and movie, where it appears.*

Thank you for this point, which was not properly explained. We are working with mTmG mice, which express dTomato in all body cells, particularly stroma. GFP is only expressed

after Cre expression. An explanation of the system has been added to the figure legend of suppl. Fig. 1.

- *3. line 141-3: the statement that Cxcr5 and Cxcr4 expression is downregulated is mentioned twice in the same sentence.*

Thank you. The second statement has been removed.